# An electronic neuromorphic system for real-time detection of high frequency oscillations (HFO) in intracranial EEG

Mohammadali Sharifshazileh[1,2,3], Karla Burelo [1,2,3], Johannes Sarnthein [2✉] & Giacomo Indiveri [1✉]

The analysis of biomedical signals for clinical studies and therapeutic applications can benefit from embedded devices that can process these signals locally and in real-time. An example is the analysis of intracranial EEG (iEEG) from epilepsy patients for the detection of High Frequency Oscillations (HFO), which are a biomarker for epileptogenic brain tissue. Mixed-signal neuromorphic circuits offer the possibility of building compact and low-power neural network processing systems that can analyze data on-line in real-time. Here we present a neuromorphic system that combines a neural recording headstage with a spiking neural network (SNN) processing core on the same die for processing iEEG, and show how it can reliably detect HFO, thereby achieving state-of-the-art accuracy, sensitivity, and specificity. This is a first feasibility study towards identifying relevant features in iEEG in real-time using mixed-signal neuromorphic computing technologies.

[1] Institute of Neuroinformatics, University of Zurich and ETH Zurich, Zurich, Switzerland. [2] Department of Neurosurgery, University Hospital Zurich, University of Zurich, Zurich, Switzerland. [3] These authors contributed equally: Mohammadali Sharifshazileh, Karla Burelo. ✉email: johannes.sarnthein@usz.ch; giacomo@ini.uzh.ch

The amount and type of sensory data that can be recorded are continuously increasing due to the recent progress in microelectronic technology[1]. This data deluge calls for the development of low-power embedded edge computing technologies that can process the signals being measured locally, without requiring bulky computers or the need for internet connection and cloud servers. A promising approach that has been recently proposed to address these challenges is the one of "neuromorphic computing"[2]. Several innovative neuromorphic computing devices have been developed to carry out computation "at the edge"[3–8]. These are general purpose brain-inspired architectures that support the implementation of spiking and rate-based neural networks for solving a wide range of spatio-temporal pattern recognition problems[9,10]. Their in-memory computing spike-based processing nature offers a low-power and low-latency solution for simulating neural networks that overcome some of the problems that affect conventional Central Processing Unit (CPU) and Graphical Processing Unit (GPU) "von Neumann" architectures[11,12].

In this paper we take this approach to the extreme and propose a very "specific purpose" neuromorphic system for bio-signal processing applications that integrates a neural recording headstage directly with the SNN processing cores on the same die, and that uses mixed-signal subthreshold-analog and asynchronous-digital circuits in the SNN cores which directly emulate the physics of real neurons to implement faithful models of neural dynamics[13,14]. Besides not being able to solve a broad range of pattern recognition tasks by design, the use of subthreshold analog circuits renders the design of the neural network more challenging in terms of robustness and classification accuracy. Nonetheless, successful examples of small-scale neuromorphic systems have been recently proposed to process bio-signals, such as Electrocardiogram (ECG) or Electromyography (EMG) signals, following this approach[15–18]. However, these systems were suboptimal, as they required external biosignal recording, frontend devices, and data conversion interfaces. Bio-signal recording headstages typically comprise analog circuits to amplify and filter the signals being measured and can be highly diverse in specification depending on the application[19]. For example, neural recording headstages for experimental neuroscience target high-density recordings[20–23] and minimize the circuit area requirements, while devices used for clinical studies and therapeutic applications require a small number of recording channels and the highest possible signal-to-noise ratio (SNR)[24–27].

The system we propose is targeted toward the construction of a compact and low-power long-term epilepsy monitoring device that can be used to support the solution of a clinically relevant problem: Epilepsy is the most common severe neurological disease. In about one-third of patients, seizures cannot be controlled by medication. Selected patients with focal epilepsy can achieve seizure freedom if the epileptogenic zone (EZ), which is the brain volume generating the seizures, is correctly identified and surgically removed in its entirety. Presurgical and intraoperative measurement of iEEG signals is often needed to identify the EZ precisely[28]. High Frequency Oscillations (HFO) have been proposed as a new biomarker in iEEG to delineate the EZ[25–27,29–36]. While HFO have been historically divided into "ripples" (80–250 Hz) and "fast ripples" (FR, 250–500 Hz), detection of their co-occurrence was shown to enable the optimal prediction of postsurgical seizure freedom[31]. In that study, HFO were detected automatically by a software algorithm that matched the morphology of the HFO to a predefined template (Morphology Detector)[31,37]. An example of such an HFO is shown in Fig. 1a. While software algorithms are used for detecting HFO offline[30,38], compact embedded neuromorphic systems that can record iEEGs and detect HFO online in real time, would be able

to provide valuable information during surgery, and simplify the collection of statistics in long-term epilepsy monitoring[39–41]. Here we show how a simple model of a two-layer SNN that uses biologically plausible dynamics in its neuron and synapse equations, can be mapped onto the neuromorphic hardware proposed and applied to real-time online detection of HFO[42]. We first describe the design principles of the HFO detection architecture and its neuromorphic circuit implementation. We discuss the characteristics of the circuit blocks proposed and present the experimental results measured from the fabricated device. We then show how the neuromorphic system performs in HFO detection compared to the Morphology Detector[31] on iEEG recorded from the medial temporal lobe[43].

## Results

Figure 1 shows how prerecorded iEEG[43] was processed by the frontend headstage and the SNN multi-core neuromorphic architecture. Signals were band-passed filtered into Ripple and Fast Ripple bands (Fig. 1a, b, f). The resulting waveforms were converted into spikes using asynchronous delta modulator (ADM) circuits[44,45] (Fig. 1c, f) and fed into the SNN architecture (Fig. 1d, g). Neuronal spiking signals the detection of an HFO (Fig. 1e bottom).

All stages were first simulated in software to find the optimal parameters and then validated with the hardware components. The HFO detection was validated by comparing the HFO rate across recording intervals (Fig. 1i) and with postsurgical seizure outcome[31].

**The neuromorphic system.** An overview of the hardware neuromorphic system components is depicted in Fig. 2. The chip (Fig. 2c) was fabricated using a standard 180 nm Complementary Metal-Oxide-Semiconductor (CMOS) process. It comprises 8 input channels (headstages) responsible for the neural recording operation, band-pass filtering and conversion to spikes[46], and a multicore neuromorphic processor with 4 neurosynaptic cores of 256 neurons each, which is a Dynamic Neuromorphic Asynchronous Processor (DYNAP) based on the DYNAP-SE device[47]. The total chip area is 99 mm². The 8 headstages occupy 1.42 mm² with a single headstage occupying an area of 0.15 mm² (see Fig. 2a). The area of the four SNN cores is 77.2 mm² with a single SNN core occupying 15 mm². For the HFO detection task, the total average power consumption of the chip at the standard 1.8 V supply voltage was 614.3 μW. The total static power consumption of a single headstage was 7.3 μW. The conversion of filtered waveforms to spikes by the ADMs consumed on average 109.17 μW. The power required by the SNN synaptic circuits to process the spike rates produced by the ADMs was 497.82 μW, while the power required by the neurons in the second layer of the SNN to produce the output spike rates was 0.2 nW. The block diagram of the hardware system functional modules is shown in Fig. 2b. The Field Programmable Gate Array (FPGA) block on the right of the figure represents a prototyping device that is used only for characterizing the system performance. Figure 2c shows the chip photograph, and Fig. 2d represents a rendering of the prototyping Printed Circuit Board (PCB) used to host the chip.

Figure 3 shows the details of the main circuits used in a single channel of the input headstage. In particular, the schematic diagram of the Low-Noise Amplifier (LNA) is shown in Fig. 3a. It consists of an Operational Transconductance Amplifier (OTA) with variable input Metal Insulator Metal (MIM) capacitors, $C_{in}$ that can be set to 2/8/14/20 pF and a Resistor-Capacitor (RC) feedback in which the resistive elements are implemented using MOS-bipolar structures[48]. The MOS-bipolar pseudoresistors $MPR1$ and $MPR2$, and the capacitors $C_f = 200$ fF of Fig. 3a were

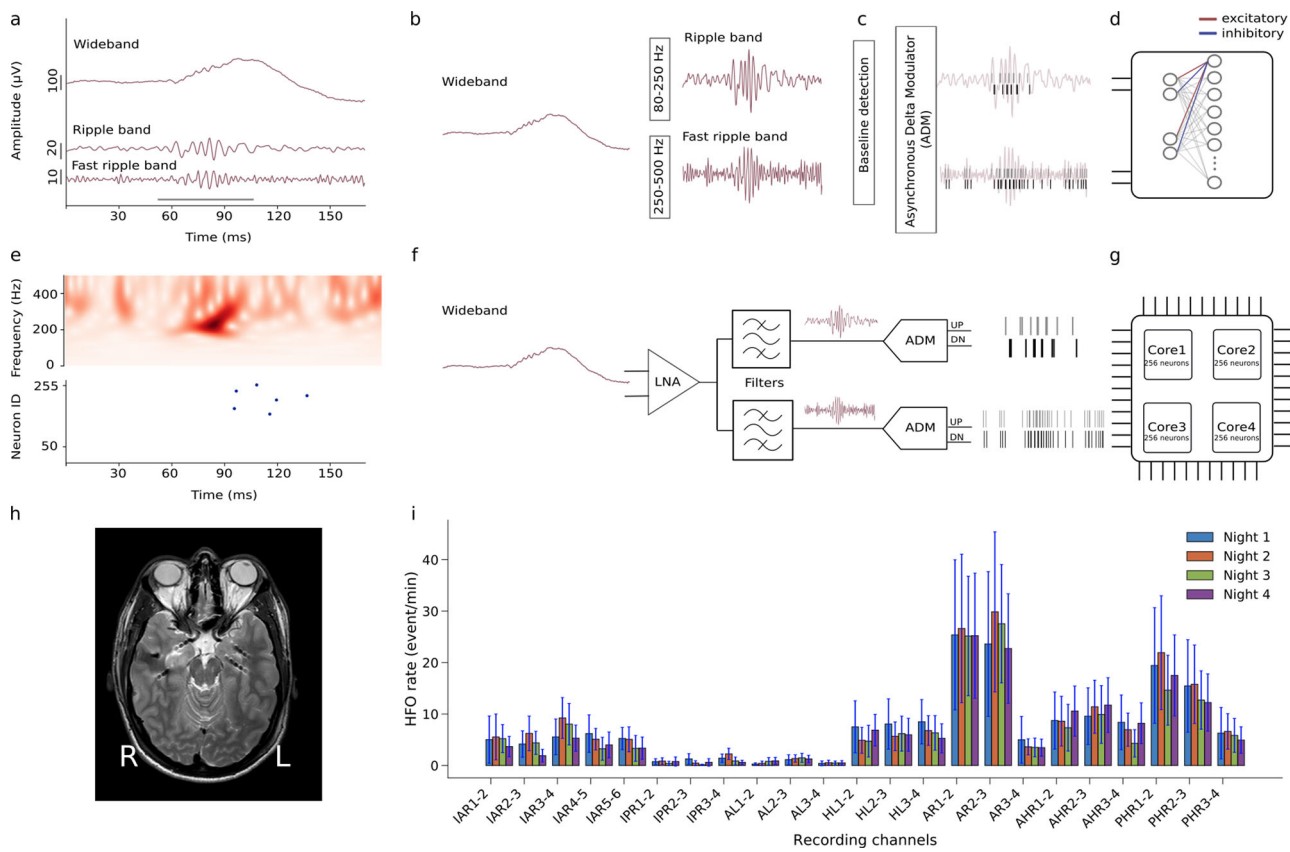

**Fig. 1 Automatic HFO detection using a bio-inspired SNN. a** The pre-recorded iEEG signal in wideband, Ripple band (80–250 Hz) and Fast Ripple band (250–500 Hz). HFO stand out of the baseline in the signal. The period marked by the gray bar represents a clinically relevant HFO[27,31]. **b–d** Software simulated spiking neural network (SNN). For preprocessing, the wideband EEG is filtered in Ripple band and Fast Ripple band. A baseline detection stage finds the optimum threshold that is applied in an Asynchronous Delta Modulator (ADM) which converts the signal to spikes. Signal traces are encoded by UP spikes (gray bars) and DOWN spikes (black bars), which are then fed as input into the SNN. The SNN is implemented as a two-layer spiking network of integrate and fire neurons with dynamic synapses. Each neuron in the second layer receives four inputs: two excitatory spike trains from UP channels and two inhibitory ones from DOWN channels. The parameters of the network were chosen to exhibit the relevant temporal dynamics and tune the neurons to produce output spikes in response to input spike train patterns that encode clinically relevant HFO. **e**, top Time-frequency spectrum of the Fast Ripple iEEG of panel **a**. **e**, bottom Firing of SNN neurons indicates the occurrence of the HFO. **f** Block diagram of the neuromorphic system input headstage. The headstage comprises a low noise amplifier (LNA), two configurable bandpass filters, and two ADM circuits that convert the analog waveforms into spike trains. **g** The spikes produced by the ADMs are sent to a multi-core array of silicon neurons that are configured to implement the desired SNN. **h** MRI with 7 implanted depth electrodes that sample the mesial temporal structures of a patient with drug-resistant temporal lobe epilepsy (Patient 1). **i** Rates of HFO detected by the neuromorphic SNN for recordings made across four nights for Patient 1. The variability of the HFO rates across intervals within a night is indicated by standard error bars. Recording channels AR1-2 and AR2-3 in the right amygdala showed the highest HFO rates which were stable over nights. Thus, the neuromorphic system would predict that a therapeutic resection, which should include the right amygdala, would achieve seizure freedom. Indeed, a resection including the right amygdala achieved seizure freedom for >1 year.

chosen to implement a high-pass filter with a low cutoff frequency of 0.9 Hz. Similarly, the input capacitors $C_{in}$, $C_f$, and the transistors of the OTA were sized to produce maximum amplifier gain of approximately 40.2 dB but can be adjusted to smaller values by changing the capacitance of $C_{in}$.

Figure 3b shows the schematic of the OTA, which is a modified version of a standard folded-cascode topology[49]. The currents of the transistors in the folded branch (M5–M10) are scaled to approximately 1/6th of the currents in the original branch M1–M4. The noise generated by M5–M10 is negligible compare to that of M1–M4 due to the low current in these transistors. As a result, the total current and the total input-referred noise of the OTA was minimized.

To ensure accurate bias-current scaling, the currents of Mb2 and Mb4 in Fig. 3b were set using the bias circuit formed by Mb1, Mb3, and Mb5. The voltages $Vb1$ and $Vb2$ in the biasing circuit can be set by a programmable 6526-level integrated parameter-generator, integrated on chip[50]. The current sources formed by

Mb1 and Mb2 were cascoded to increase their output impedance and to ensure accurate current scaling. These devices operate in strong inversion to reduce the effect of threshold voltage variations. The source-degenerated current mirrors formed by M11, M12, Mb5, and resistors R1 and $k \times$ R1 assure that the currents in M5 and M6 are a small fraction of the currents in M3 and M4. The R1 gain coefficient $k$ was chosen at design time to be $k = 8.5$. Thanks to the use of this source-degenerated current source scheme, the $1/f$ noise in the OTA is limited mainly to the effect of the input differential pair. Therefore, the transistors of the input-differential pair were chosen to be pMOS devices and to have a large area.

The active filters implemented in our system are depicted in Fig. 3c. They comprise three operational amplifiers, configured to form a Tow-Thomas resonating architecture[51]. This architecture consists of a damped inverting integrator that is stabilized by $R2$ and cascaded with another undamped integrator, and an inverting amplifier for adjusting the loop-gain by a factor set by

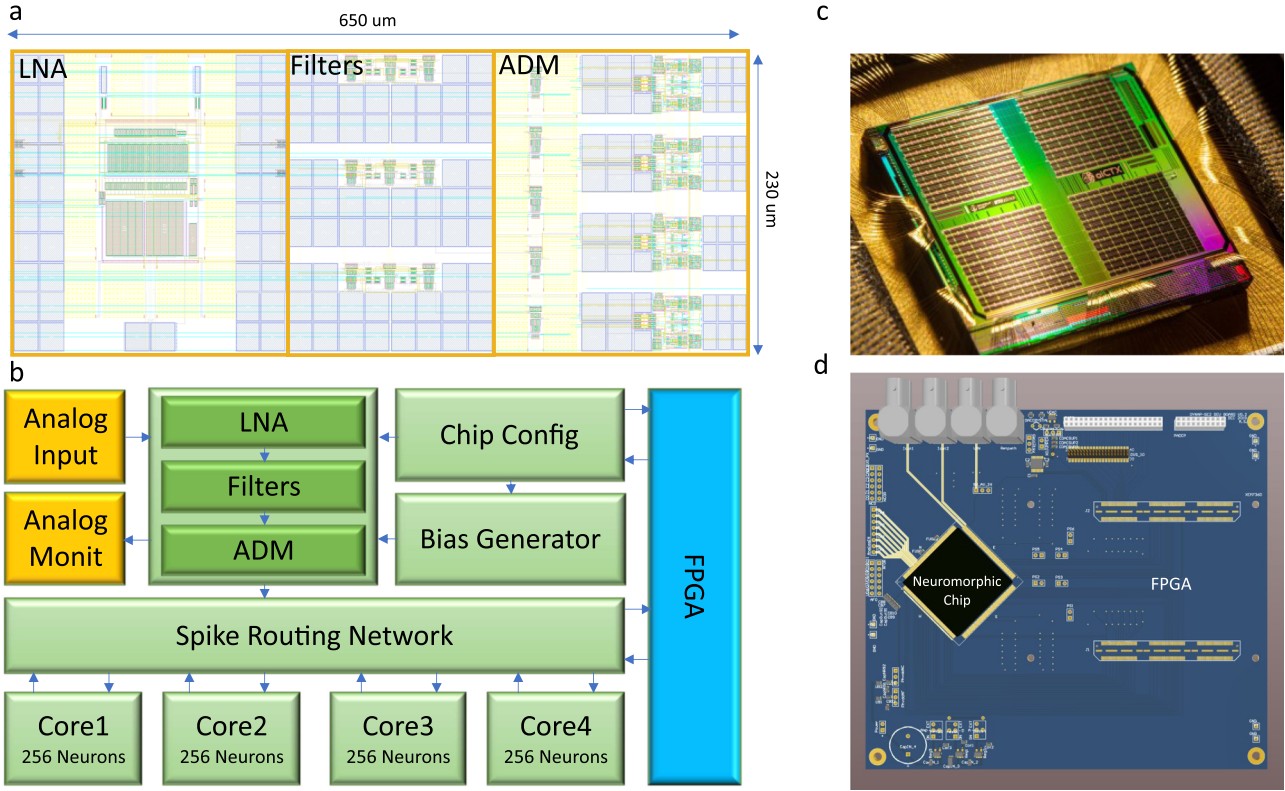

**Fig. 2 Neuromorphic electronic system overview. a** Physical layout of a single channel of the analog headstage array, including the LNA, three low-pass/band-pass filters, and four ADM signal to spike encoders. **b** Reduced block diagram of the neuromorphic platform. Analog signals from electrodes are fed into the input headstage that converts them into spike trains and sends them to the SNN implemented on the multi-neuron cores, via a spike routing network. The spike routing network routes the spikes within on-chip SNN and to an external FPGA device used for data logging and prototyping. The FPGA is also used for setting circuit parameters. **c** Chip photograph showing the 11 mm x 9 mm silicon die. **d** Prototyping Printed Circuit Board (PCB) used to host the chip and the infrastructure to implement the test setup. The setup is composed of a prototyping FPGA board mounted on the same PCB that hosts the chip, and of probe points to evaluate the characteristics of both input headstage and SNN multi-core network.

the ratio $R6/R5$. The center frequency $f_0$ of the bandpass filter can be calculated as $f_0 = 1/2\pi\sqrt{R3R4C1^2}$. By choosing $R3 = R4 = R$, we can then simplify it to $f_0 = 1/2\pi RC1$. Similarly, the gain of the filter is $|T_{BP}| = R4/R1$ and its bandwidth $BW = 2\pi f_0\sqrt{R3R4}/R2$, but with our choice of resistors we can show that $|T_{BP}| = R/R1$ and $BW = 1/(R2C1)$. Therefore, this analysis shows that $R$ is responsible for setting $f_0$, $R1$ for adjusting the gain, and $R2$ for tuning the bandwidth. Moreover, due to the resistive range of the tunable double-PMOS pseudo resistors used in this design[52], $f_0$ was set in the sub-hundred Hertz region by choosing $C1 = 10 \, pF$.

Figure 3d shows the schematic diagram of the ADM circuit[45]. There are four such circuits per headstage channel. One for converting the wideband signal $V_{amp}$ into spike trains; one for converting the output of the low-pass filter $V_{out\_lowpass}$; and two for converting the output of the two band-pass filters $V_{out\_bandpass}$. The amplifier at the input of the ADM circuit in Fig. 3d implements an adaptive feedback amplification stage with a gain set by $C_{in}/C_f$ that in our design is equal to 8 when $V_{reset}$ is high, and approximately zero during periods in which $V_{reset}$ is low. In these periods, defined as "reset assertion" the output of the amplifier $V_e$ is clamped to $V_{ref}$, while in periods when $V_{reset}$ is high, the output voltage $V_e$ represents the amplified version of the input. The $V_e$ signal is then sent as input to a pair of comparators that produce either "UP" or "DN" digital pulses depending if $V_e$ is greater than $V_{tu}$ or lower than $V_{td}$. These parameters set the ADM circuit sensitivity to the amplitude of the Delta-change. The smallest values that these voltages can take is limited by the input

offset of the ADM comparators (see $CmpU, CmpD$ in Fig. 3d), which is approximately 500 $\mu$V.

Functionally, the ADM represents a Delta-modulator that quantizes the difference between the current amplitude of $V_e$ and the amplitude of $V_e$ at the previous reset assertion. The precise timing of the UP/DN spikes produced in this way is deemed to contain all the information about the original input signal, given that the parameters of the ADM are known[53]. The UP and DN spikes are used as the request signals of the asynchronous AER communication protocol[54–56] used by the spike routing network for transmitting the spikes to the silicon neurons of the neuromorphic cores (Fig. 2). We call this event-based computation. These signals are pipelined through asynchronous buffers that locally generates $ACK_{UP/DN}$ to reset the ADM with every occurrence of an UP or DN event. The output of the asynchronous buffer $REQ_{UP/DN(toSNN)}$ conveys these events to the next asynchronous stages. The bias voltage $V_{refr}$ controls the refractory period that keeps the amplifier reset and limits the maximum event rate of the circuit to reduce power consumption. The bias voltages $V_{tu}$ and $V_{td}$ control the sensitivity of the ADM and the number of spikes produced per second, with smaller values producing spike trains with higher frequencies. Small $V_{tu}$ and $V_{td}$ settings lead to higher power consumption and allow the faithful reconstruction of the input signal with all its frequency components. The ADM hyper parameters $V_{refr}$, $V_{tu}$, and $V_{td}$ can therefore be optimized to achieve high reconstruction accuracy of the input signal and suppress background noise (e.g., due to high-frequency signal components), depending on the nature of the signal being processed (see Methods). All of the 32 ADM output

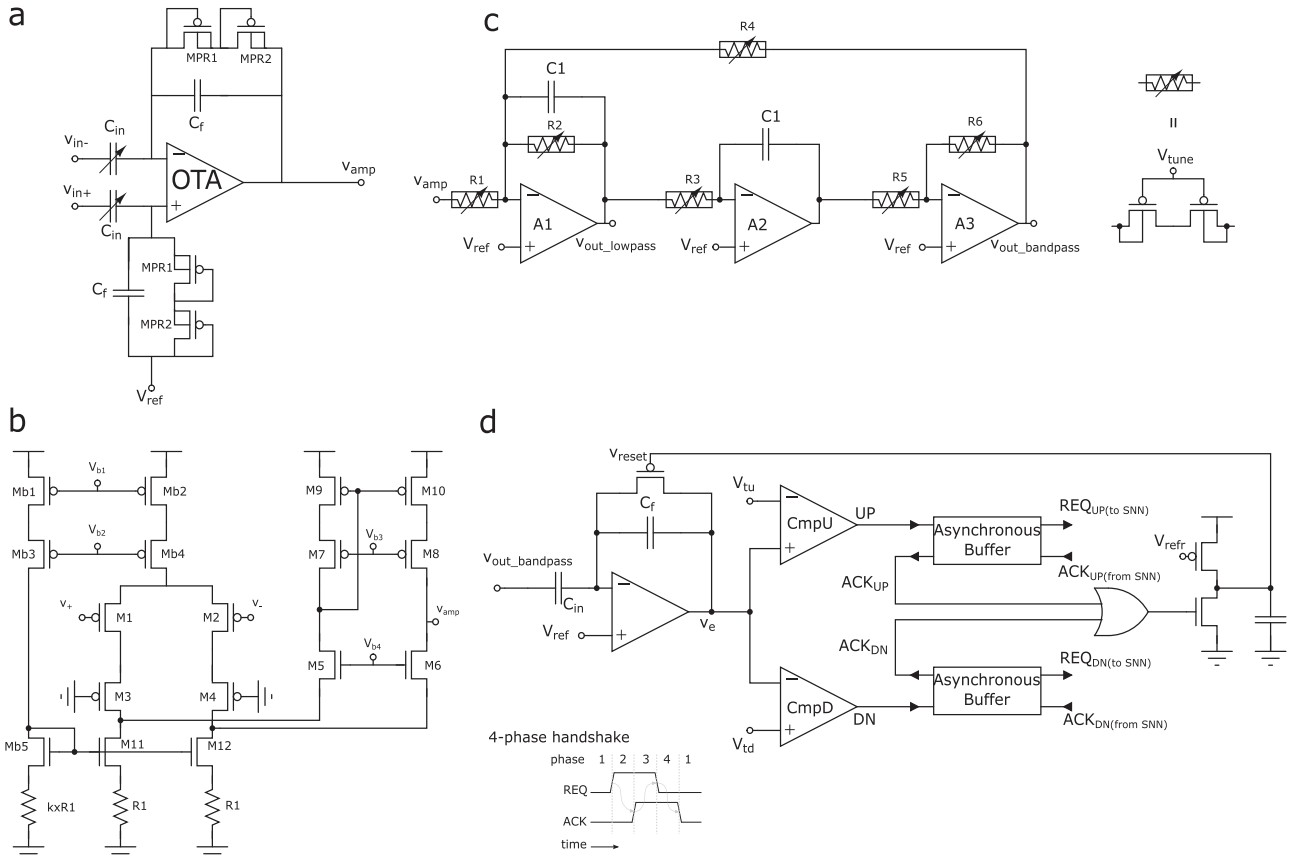

**Fig. 3 Schematic diagrams of the input headstage circuits. a** Variable-gain LNA using variable input capacitor array and pseudo-resistors. The gain of this stage is calculated by $C_{in}/C_f$; the use of the pseudo-resistors allows to reach small low cut-off frequencies. **b** Folded cascode OTA using resistive degeneration to reduce the noise influence of nMOS devices. Note that the current flowing through the biasing branch, $Mb1\text{-}Mb3\text{-}Mb5\text{-}k \times R1$, is $k$ times smaller than the tail branch of the amplifier. **c** band-pass (Tow-Thomas) filters for performing second-order filtering in both ripple and fast-ripple bands as well as the low-frequency component of the iEEG. Tunable pseudo resistors are used to adjust the filter gain, center-frequency, and band-width. The same basic structure can be used to provide both low-pass and band-pass outputs, thus is desirable in terms of design flexibility. **d** Asynchronous Delta Modulator (ADM) circuit to convert the analog filter outputs into spike trains. The ADM input amplifier has a gain of $C_{in}/C_f$ in normal operation when $V_{reset}$ is low and the feedback PMOS switch is off. As the amplified signal crosses one of the two thresholds, $V_{tu}$ or $V_{td}$, a UP or DN spike is produced by asserting the corresponding REQ signal. A 4-phase handshaking mechanism produces the corresponding ACK signal in response to the spike. Upon receiving the ACK signal, the ADM resets the amplifier input and goes back to normal operation after a refractory period determined by the value of $V_{refr}$. The asynchronous buffers act as 4-phase handshaking interfaces that propagate the UP/DN signals to the on-chip AER spike routing network of Fig. 2.

channels are then connected to a common AER encoder which includes an AER arbiter[57] used to manage the asynchronous traffic of events and convey them to the on-chip spike routing network.

**Analog headstage circuit measurements**. Figure 4 shows experimental results measured from the different circuits present in the input headstage. Figure 4a shows the transient response of the LNA to a prerecorded iEEG signal used as input. The signal was provided to the headstage directly via an arbitrary waveform generator programmed with unity gain and loaded with a sequence of the prerecorded iEEG data with amplitude in the mV range. We also tested the LNA with an input sine wave of 100 Hz with a 1 mV peak-to-peak swing, revealing <1% of total harmonic distortion at maximum gain, and an output swing ranging between 0.7 V and 1.4 V. To characterize the input referred noise of the LNA, we shorted input terminals of the LNA, $V_{in+}$, $V_{in-}$, captured LNA output, $V_{amp}$, on a dynamic signal analyzer and divided it by the gain of the LNA, set to 100, and plotted output power spectral density (see Fig. 4b). The LNA generates a ≤100 nV/$\sqrt{Hz}$ noise throughout the spectrum. As the 1/$f$ noise dissipates with the increase in frequency, the LNA noise only scales

down to the thermal component. Thus, the noise for the Ripple band is < 10 nV/$\sqrt{Hz}$ and <5 nV/$\sqrt{Hz}$ for the Fast Ripple band. Figure 4b shows how the noise generated by the LNA is well below the pre-recorded iEEG power, throughout the full frequency spectrum.

The LNA features a programmable gain that can be set to 20 dB, 32 dB, 36 dB, or 40 dB. It has a > 40 dB common mode rejection ratio;

it consumes 3 µW of power per channel; and it has a total bandwidth, defined as $Gm/(A_M \cdot C_L)$, approximately equal to 11.1 KHz, when the capacitive load is set to $C_L$=20 fF, the OTA transconductance to $Gm$=20 nS and the amplifier gain to 40 dB (see Fig. 4c).

The on-chip parameter generator can be used to set the filter frequency bands. For HFO detection, we biased the tunable pseudoresistors of the filters to achieve a cutoff frequency of 80 Hz for the low-pass filter, a range between 80 Hz and 250 Hz for the first bandpass filter, appropriate for Ripple detection, and between 250 Hz and 500 Hz for the second bandpass filter, to detect Fast Ripples (see Fig. 4c).

As we set the tail current of each single-stage OpAmp of Fig. 3c to 150 nA, each filter consumed 0.9 µW of power.

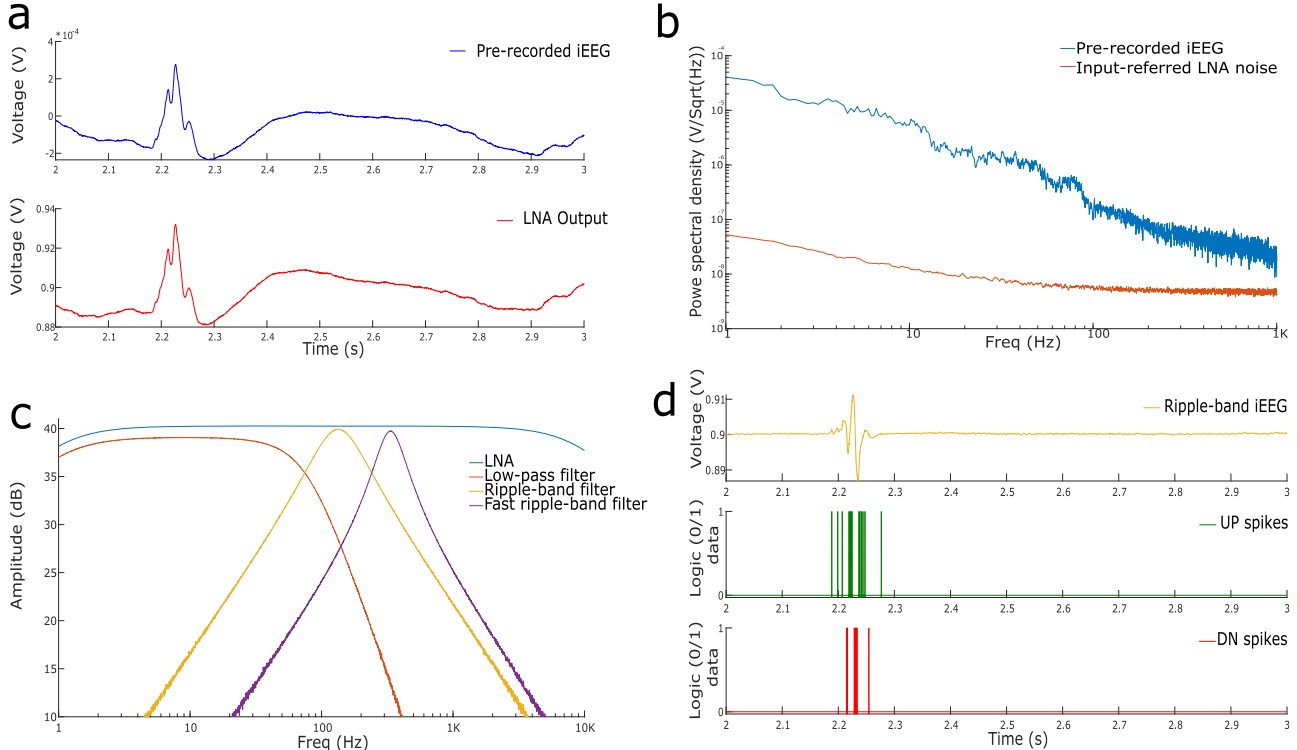

**Fig. 4 Measurements from the analog headstage. a** An iEEG sample[43] and the LNA amplified output. **b** Noise floor of the headstage LNA and iEEG power spectral density. In the HFO range (80–250 Hz) the noise level of the LNA is below the iEEG noise floor by an order of magnitude. **c** Frequency response of the implemented filters in the headstage. The band-pass filters are tuned to highlight HFO frequency bands. **d** ADM circuit simulation response to the ripple-filtered signal. The top plot shows the analog filter output, the middle plot shows the UP spikes generated by the ADM and the bottom plot shows the DN spikes.

Figure 4d plots the spikes produced by the AMD circuit in response to Ripple-band data obtained from the pre-recorded iEEG measurements. We set the ADM refractory period to 300 μs, making it the longest delay, compared to those introduced by the comparator and handshaking circuits, that are typically < 100 μs. The spike-rate of the ADM can range from few hundreds of Hz to hundreds of kHz depending on the values of $V_{tu}$, $V_{td}$, and $V_{refr}$. Each ADM consumed 1.5 nJ of energy per spike, and had a static power dissipation of 96 nW.

**System performance**. The system-level performance is assessed by measuring the ability of the proposed device to correctly measure the iEEG signals, to properly encode them with spike trains, and to detect clinically relevant HFO[31] via the SNN architecture.

The SNN architecture is a two-layer feed-forward network of integrate and fire neurons with dynamic synapses, i.e., synapses that produce post-synaptic currents that decay over time with first-order dynamics. The first layer of the network comprises four input neurons: the first neuron conveys the UP spikes that encode the iEEG signal filtered in Ripple band; the second neuron conveys the DN spikes derived from the same signal; the third neuron conveys the UP spikes derived from the Fast-ripple band signal, while the forth neuron conveys the DN spikes. The second layer of the network contains 256 neurons which receive spikes from all neurons of the input layer (see Fig. 5b). The current produced by the dynamic synapses of the second layer neurons decay exponentially over time at a rate set by a synapse-specific time-constant parameter. The amplitude of these currents in response to the input spikes depends on a separate weight parameter, and their polarity (positive or negative) depends on

the synapse type (excitatory or inhibitory). In the network proposed, UP spikes are always sent to excitatory synapses and DN spikes to inhibitory ones. All neurons in the second layer have the same connectivity pattern as depicted in Fig. 5b with homogeneous weight values. An important aspect of the SNN network lies in the way it was configured to recognize the desired input spatio-temporal patterns: rather than following the classical Artificial Neural Network (ANN) approach of training the network by modifying the synaptic weights with a learning algorithm and using identical settings for all other parameters of synapses and neurons, we fixed the weights to constant values and chose appropriate sets of parameter distributions for the synaptic time constants and neuron leak variables. Because of the different time-constants for synapses and neurons, the neurons of the second layer produce different outputs, even though they all receive the same input spike trains.

The set of parameter distributions that maximized the network's HFO detection abilities was found heuristically by analyzing the temporal characteristics of the input spike trains and choosing the relevant range of excitatory and inhibitory synapse time constants that produced spikes in the second layer only for the input signals that contained HFO as marked by the Morphology Detector[31] (see Figs. 1a, 5a). This procedure was first done using software simulations with random number generators and then validated in the neuromorphic analog circuits, exploiting their device mismatch properties.

The software simulations were carried out using a behavioral-level simulation toolbox based on the neuromorphic circuit equations, that accounts for the properties of the mixed-signal circuits in the hardware SNN[58].

The hardware validation of the network was done using a single core of the DYNAP-SE neuromorphic processor[47], which is a

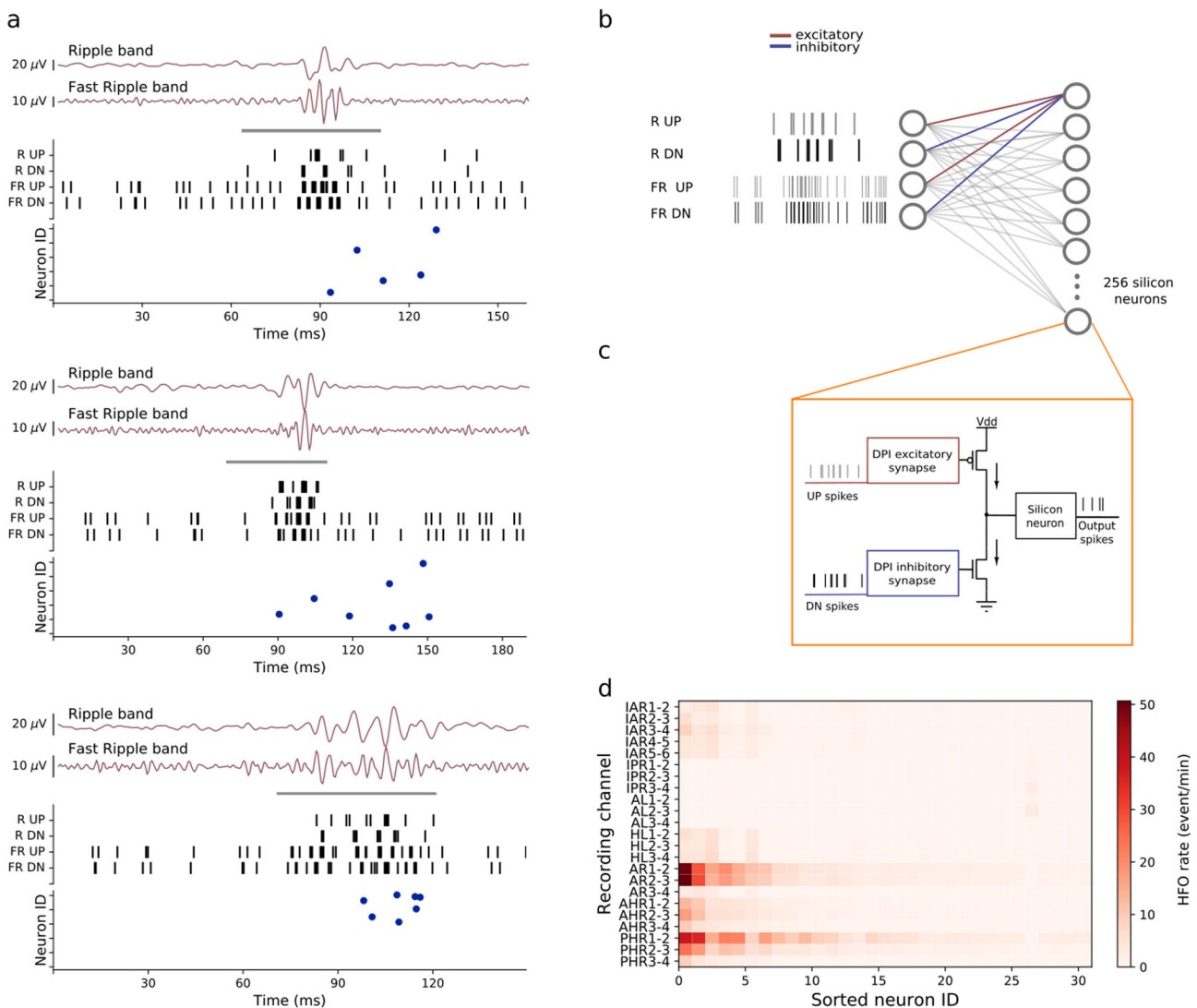

**Fig. 5 Network spiking characteristics. a** Examples of HFO that the hardware SNN detected in the iEEG of Patient 1. The periods marked by the gray bar represent clinically relevant HFO[31,43]. The signals in Ripple and Fast-ripple band were transformed to UP and DN spikes. These spike trains were sent to the neurons in the hardware through the $R_{UP}$, $R_{DN}$, $FR_{UP}$ and $FR_{DN}$ channels. The bottom panel of each example shows the raster plot of the silicon neurons. Each neuron responds to different HFO depending on the characteristics of the pattern. **b** The SNN architecture consist of a two-layer network of 256 integrate and fire neurons with dynamic synapses. Each neuron in the second layer receives excitatory connections from the $R_{UP}$ and $FR_{UP}$ channels, and inhibitory connections from the $R_{DN}$ and $FR_{DN}$ channels. The synaptic parameters time constants and weights are distributed randomly within a predetermined optimal range. **c** Hardware building blocks used for the implementations of the SNN: the DPI synapse is a "Differential-Pair Integrator" circuit[59], and the silicon neuron is an Adaptive Exponential Integrate and Fire (AdExp IF) circuit[81]. **d** HFO rates computed for Patient 1. The neurons are sorted according to their average firing rate. Only a small number of neurons fire across all the recordings, even for channels with high HFO rates (e.g., AR1-2).

previous generation chip functionally equivalent to the one proposed, implemented using the same CMOS 180 nm technology node. The 256 neurons in this core received spikes produced by the ADM circuits of the analog headstage, as described in Fig. 5c. The ADM UP spikes were sent to the excitatory synapses, implemented using a DPI circuit[13,59] that produce positive currents, and DN spikes were sent to complementary versions of the circuit that produce inhibitory synaptic currents. Both excitatory and inhibitory currents were summed into the input nodes of their afferent leaky integrate-and-fire silicon neuron circuit, which produced output spikes only if both the frequency and the timing of the input spikes was appropriate (see Fig. 5c). The bias values of the excitatory and inhibitory DPI circuits and of the neuron leak circuits were set in a way to match the mean values of the software simulation parameters. All neuron and synapse circuits in the same core of the chip share global bias parameters, so nominally all excitatory

synapses would have the same time-constant, all inhibitory ones would share a common inhibitory time-constant value and all neurons would share the same leak parameter value. However, as the mixed-signal analog-digital circuits that implement them are affected by device mismatch, they exhibit naturally a diversity of behaviors that implements the desired variability of responses. Therefore, in the hardware implementation of the SNN, the distribution of parameters that produce the desired different behaviors in the second layer neurons emerges naturally, by harnessing the device mismatch effects of the analog circuits used and without having to use dedicated random number generators[60,61]. Analysis of the data presented in Fig. 5 shows that an average number of 64 neurons were sufficient for detecting an HFO from a single channel input.

Figure 1e (bottom panel) shows an example of the activity of the hardware SNN in response to an HFO that was labeled as

**Table 1 Patient characteristics and postsurgical seizure outcome. We "predict" seizure outcome for each patient based on resection of the HFO area that was delineated by the Morphology Detector[31] and the hardware SNN of our system. The hardware SNN prediction was correct in 7/9 patients.**

| Patient | Histology/ Pathology | Intervals of 5 min | Test-retest intervals | Outcome (ILAE) | Follow-up (months) | Morphology Detector prediction | Hardware SNN prediction |
|---|---|---|---|---|---|---|---|
| 1 | HS | 28 | 0.95 | 1 | 12 | TN | TN |
| 2 | Glioma | 13 | 0.97 | 1 | 29 | TN | TN |
| 3 | HS | 39 | 0.83 | 1 | 13 | TN | TN |
| 4 | HS | 34 | 0.96 | 1 | 41 | TN | TN |
| 5 | HS | 35 | 0.91 | 1 | 14 | TN | TN |
| 6 | HS | 35 | 0.59 | 1 | 11 | TN | TN |
| 7 | HS | 1 | — | 3 | 42 | FN | FN |
| 8 | HS | 16 | 0.74 | 3 | 15 | FN | TP |
| 9 | HS | 12 | 0.90 | 5 | 46 | FN | FN |

HS hippocampal sclerosis; ILAE seizure outcome classification of the International League Against Epilepsy.

clinically relevant by the Morphology Detector[31]. The iEEG traces in the Ripple band and Fast ripple band (Fig. 1a) and the time frequency spectrum (top panel of Fig. 1e) show the HFO shortly before the SNN neurons spike in response to it (bottom panel of Fig. 1e). The delay between the beginning of the HFO and the spiking response of the silicon neurons is due to the integration time of both excitatory and inhibitory synapse circuits, which need to accumulate enough evidence for producing enough positive current to trigger the neuron to spike.

To improve classification accuracy and robustness, we adopted an ensemble technique[62] by considering the response of all the 256 neurons in the network: the system is said to detect an HFO if at least one neuron in the second layer spikes in a 15 ms interval. We counted the number of HFO detected per electrode channel and computed the corresponding HFO rate (Section 4). Examples of HFO recorded from Patient 1 and detected by the hardware SNN are shown in Fig. 5a; several neurons respond within a few milliseconds after initiation of the HFO. Different HFO produce different UP and DN spike trains, which in turn lead to different sets of second layer neurons spiking. Figure 5d shows the HFO rates calculated for each electrode from the recordings of this patient. Observe that not all the neurons in the second layer respond to the HFO. Even for electrode channels with high HFO rates, a very small number of neurons fire at high rates.

The robustness of the HFO rate measured with our system can be observed in Fig. 1i, where the relative differences of HFO rates across channels in Patient 1 persisted over multiple nights. To quantify this result we performed a test-retest reliability analysis by computing the scalar product of the HFO rates across all recording intervals (0.95 in Patient 1), where the scalar product is ~Ĩ for highly overlapping spatial distributions, indicating that the HFO distribution persists over intervals.

**Predicting seizure outcome.** In Patient 1, the electrodes were implanted in right frontal cortex (IAR, IPR), the left medial temporal lobe (AL, HL) and the right medial temporal lobe (AR, AHR, PHR). The recording channels AR1-2 and AR2-3 in the right amygdala produced the highest HFO rates persistently. We included all channels with persistently high HFO rate in the 95% percentile to define the "HFO area". If the HFO area is included in the resection volume (RV), we would retrospectively "predict" for the patient to achieve seizure freedom. Indeed, right selective amygdalohippocampectomy in this patient achieved seizure freedom for >1 year.

We validated the system performance across the whole patient group by performing the test-retest reliability analysis of all the data. The test-retest reliability score ranges from 0.59 to 0.97 with

a median value of 0.91. We compared the HFO area detected by our system with the RA. For each individual, we then retrospectively determined whether resection of the HFO area would have correctly "predicted" the postsurgical seizure outcome (Table 1). Seizure freedom (ILAE 1) was achieved in 6 of the 9 patients. To estimate the quality of our "prediction", we classified each patient as follows: we defined as "True Negative" (TN) a patient where the HFO area was fully located inside the RV and who became seizure free; "True Positive" (TP) a patient where the HFO area was not fully located within the RV and the patient suffered from recurrent seizures; "False Negative" (FN) a patient where the HFO area was fully located within the RV but who suffered from recurrent seizures; "False Positive" (FP) a patient where the HFO area was not fully located inside the RV but who nevertheless achieved seizure freedom.

The HFO area defined by our system was fully included in the RV in patients 1 to 6. These patients achieved seizure freedom and were therefore classified as TN. In Patients 7 and 9, the HFO area was also included in the RV but these were classified as FN since these patients did not achieve seizure freedom. The false prediction may stem either from HFO being insufficiently detected or from the epileptogenic zone being insufficiently covered by iEEG electrode contacts. In Patient 8, the HFO area was not included in the RV and the patient did not achieve seizure freedom (TP).

We finally compared the predictive power of our detector to that of the Morphology Detector[31] for the individual patients (Table 1) and over the group of patients (Table 2). The overall prediction accuracy of our system across the 9 patients is comparable to that obtained by the Morphology Detector. The 100% specificity achieved by both detectors indicates that HFO analysis provides results consistent with the current surgical planning.

**Discussion**

The results presented here demonstrate the potential of neuromorphic computing for "extreme-edge" use cases; i.e., computing applications for compact embedded systems that cannot rely on internet or "cloud computing" solutions. It should be clear however, that this approach does not address general purpose neuromorphic computing classes of problems nor does it propose novel methodologies for artificial intelligence applications. It is an approach that needs to be optimized to every individual "specific purpose" use case, by reducing to the minimum necessary the amount of compute resources, to minimize power consumption. As such, this approach can not be directly compared with large-scale neuromorphic computing approaches, or state-of-the-art or deep-learning methods.

**Table 2 Comparison of postsurgical outcome prediction between the Morphology Detector and our system. TP True Positive; TN True Negative; FP False Positive; FN False Negative; N = TP + TN + FP + FN = number of patients.**

|  | Morphology detector prediction [%] | Hardware SNN prediction [%] |
|---|---|---|
| **Specificity** = TN/(TN + FP) | 100 | 100 |
| **Sensitivity** = TP/(TP + FN) | 0 | 33 |
| **Negative Predictive Value** = TN/(TN + FN) | 67 | 75 |
| **Positive Predictive Value** = TP/(TP + FP) | — | 100 |
| **Accuracy** = (TP + TN)/N [%] | 67 | 78 |

The Morphology Detector did not classify a TP so that sensitivity and PPV can not be calculated.

Other embedded systems and VLSI devices designed for the specific case of processing and/or classifying EEG signals have been proposed in recent years[63–66]. Table 3 highlights the differences between these systems and the one presented in this work. Interestingly, only two of these other designs have opted for integrating analog acquisition headstages with the computing stages for standalone operation[65,66]. Both of these designs have a comparable number of channels to our system; however they comprise conventional analog to digital converter designs (ADCs) that are not optimal for processing bio-signals[67]. Indeed analog to digital data conversion for bio-signal processing has been an active area of investigation in biomedical processing field, with increasing evidence in favor of delta encoding schemes (such as the one used in this work)[68–70]. The only design listed in Table 3 that utilizes a symbolic encoding scheme and that has been applied to iEEG is the work of Burello et al.'19[63]. However, this design is missing a co-integrated analog headstage and, by extension, an integrated local binary pattern encoder. Separating the signal encoding stage from the processing stage allows the implementation of sophisticated signal processing techniques and machine learning algorithms, as is evident from the works of Burello et al.[63] and Feng et al.[64]. But using off-the shelf platforms for signal encoding, processing, or both, leads to much higher power consumption and bulky platforms that make the design of compact and portable embedded systems more challenging.

Other full custom and low-power neural recording headstages developed in the past were optimized for very large scale arrays[20–23], or for intracranial recordings[49,71–76]. To our knowledge, we present here the first instance of a headstage design that has the capability of adapting to numerous use cases requiring different gain factors and band selections, on the same input channel.

When comparing our neuromorphic SNN with other HFO detectors proposed in the literature[35,36], several differences and commonalities become evident. As a conceptual difference, the SNN proposed here models many of the features found in biological neural processing systems, such as the temporal dynamics of the neuron and synapse elements, or the variability in their time constants, refractory periods, and synaptic weights. Our system does not store the raw input signals by design so that off-line post-hoc examination of HFO is not possible. In addition, the approach followed to determine the right set of the model parameters is radically different from the deep-learning one: rather than using arrays of identical neurons with homogeneous parameters and a learning algorithm to determine the weights of static synapses, we tuned the parameters governing the dynamics of the synapses and exploited the variability in their outputs, using ranges that are compatible with the distributions measured from the analog circuits, to create an ensemble of weak classifiers that can reliably and robustly detect HFO. The event-based nature of the hardware implementation of such model and the matched filter properties of the SNN circuits with the time constants of the signals being processed, translates into an extremely

low-power (sub mW) device. These results demonstrate the feasibility of compact low-power implantable devices for long-term monitoring of the epilepsy severity. As a commonality to other non-neuromorphic off-line HFO detectors[25,30–33,43] (Table 1), our system uses the detection of HFO to "predict" seizure freedom after resective epilepsy surgery in individual patients.

The simulations of the SNN not only allowed us to define the optimal architecture for HFO detection, but also gave us solutions for setting the hyperparameters of the analog headstage, such as the refractory period $V_{ref}$ and the threshold ($V_{tu}$ and $V_{td}$) for the signal-to-spike conversion of the ADM. The robustness of the software simulation has now been confirmed in an independent dataset of intraoperative ECoG recordings, where the exact same simulated SNN presented here was highly successful in detecting HFO - without fine tuning of parameters at all[38]. While mismatch effect is generally a concern in modeling hardware designs based on software simulations, we show here that the mismatch among the silicon neurons resulted in a key feature for the implementation of our SNN architecture. This advantage allowed us to generate the normal distribution of parameters without manually defining the distribution of neuronal time-constants found in simulations or requiring extra memory to allocate these values. By averaging over both time and the number of neurons recruited by the ensemble technique, the SNN network was able to achieve robust results: the accuracy obtained by the SNN is compatible with that obtained by a state-of-the-art software algorithm implemented using complex algorithms on a powerful computer[31].

In the "prediction" across the patient group (Table 2), the sensitivity stands out to be very low, i.e. even after the HFO area was resected, Patients 7 and 9 suffered from recurrent seizures (FN prediction). On one hand, a FN may be due to insufficient HFO detection. On the other hand, the spatial sampling of iEEG recordings may be insufficient for localizing the EZ, i.e. seizures may originate from brain volumes where they remain undetected by the iEEG recordings. This spatial sampling restriction affects the two HFO detectors and current clinical practice all alike. Indeed, current clinical practice advised resection of brain tissue in Patients 7, 8, and 9, which nevertheless suffered from recurrent seizures postoperatively, i.e. the EZ was not removed in its entirety. Interestingly, the HFO area of the Hardware SNN might have included the correct EZ in Patient 8 (TP) while this was not the case for the Morphology detector (FN). Still, the NPV in Table 2 is the most relevant quantity for clinical purpose and it is sufficiently high for both the Morphology Detector and the Hardware SNN. Overall, the high specificity (100%) achieved with our system not only generalizes the value of the detected HFO by the SNN across different patients, but still holds true at the level of the individual patients, which is a prerequisite to guide epilepsy surgery that aims for seizure freedom.

This is a first feasibility study towards identifying relevant features in intracranial human data in real-time, on-chip, using event-based processors and spiking neural networks. By

**Table 3 Comparison of the hardware setup proposed with analogous state-of-the-art designs and methods.**

| Feature/Design | This work | Burello et al.[63] | Feng et al.[64] | Van Helleputte et al.[65] | Yoo et al.[66] |
|---|---|---|---|---|---|
| Analog Headstage | 8 ch + 32 filters | No | No | 3 ch ECG + ETI | 8 ch + 8 filters |
| Encoding | Asynchronous delta-encoding | Synchronous local binary pattern | External ADC @173.61S/s/@256S/s | Sigma-delta ADC @500S/s | Multiplexed SAR ADC @32KS/s |
| Application | HFO detection | Seizure detection | Seizure detection | Personal health monitoring | Seizure classification |
| Power consumption (headstage +processor) | 58.4 uW + 555.6 uW | 64 mW* | 45 mW*** | 183 uW + 191 uW** | 66 uW + 2.03 uJ / classification |
| Platform | Custom analog + DYNAP | Nvidia Tegra X2 | Altera Cyclone II FPGA | Custom analog+ ARM CortexM0 | Custom |
| Learning/Processing | SNN, randomized dataset | Hyperdimensional vectors, specific dataset | SVM, both randomized and specific datasets | ICA, PCA, CWT and feature extraction | SVM, specific dataset |

* Power for processing 24 channels.
** Power measured for R-peak detection with motion artefact reduction.
*** Not multichannel.

integrating on the same chip both the signal acquisition headstage and the neuromorphic multi-core processor, we developed an integrated system that can demonstrate the advantages of neuromorphic computing in clinically relevant applications. The general approach of building sensors that can convert their outputs to spikes and of interfacing spiking neural network circuits and systems on the same chip can lead to the development of a new type of "neuromorphic intelligence" sensory-processing devices for tasks that require closed-loop interaction with the environment in real-time, with low latency and low power budget. By providing "neuromorphic intelligence" to neural recording circuits the approach proposed will lead to the development of systems that can detect HFO areas directly in the operation room and improve the seizure outcome of epilepsy surgery.

## Methods

**Design and setup of the hardware device.** The CMOS circuit simulations were carried-out using the Cadence® Virtuoso ADE XL design tools. All circuits including the headstage, the parameter generator, and the silicon neurons were designed, simulated and analyzed in analog domain. The asynchronous buffers, spike routing network and chip configuration blocks were simulated and implemented in the asynchronous digital domain. The layout of the chip was designed using the Cadence® Layout XL tool. The design rule check, layout versus schematic and post-layout extraction of the analog headstages were performed using the Calibre tool. We packaged our device using a ceramic 240-pin quadratic flat package. The package was then mounted on an in-house designed six-layer printed circuit board. The programming and debugging of the System-on-Chip (SoC) was performed using low-level software and firmware developed in collaboration with SynSense Switzerland, and implemented using the XEM7360 FPGA (Opal Kelley, USA). The pre-recorded iEEG was fed to the chip using a Picoscope 2205A MSO (Picotech, UK). All frequency-domain measurements were performed using a Hewlett-Pacard 35670A dynamic signal analyzer.

**Characteristics of the SNN model.** The SNN model is composed of a layer of input units, that provide the input spikes derived from the recorded and filtered input waveforms, and a layer of Integrate-and-Fire (I&F) neurons that reproduce the dynamics of neuromorphic circuits present in the chip[13]. The silicon neurons present in the chip reproduce the properties of Adaptive-Exponential Integrate and Fire (AdEp-I&F) neuron models[77], while the synapse circuits that connect the input nodes to the AdEp-I&F neurons exhibit first-order temporal dynamics[59]. Unlike classical Artificial Neural Networks (ANNs) this model does not rely only on the synaptic weights to carry out it's task: each neuron in the network can be interpreted as a non-linear temporal filter that is tuned to the specific shape of the waveform it is trying to recognize. The tuning hyper-parameters that are relevant for this operation, besides the weights, are the neuron and synapse time constants. The equations that describe the behavior or AdExp-I&F neurons are the following:

$$\tau_{mem}\frac{d}{dt}V_{mem}(t) = -V_{mem}(t) + I_{syn}(t) - v_{ahp}(t) + f(V_{mem})$$
$$\tau_{ahp}\frac{d}{dt}v_{ahp}(t) = -v_{ahp}(t) + w_{ahp}\delta_{spk}(t)$$

(1)

where $V_{mem}$ represents the neuron's membrane potential, $f(\cdot)$ is an exponential function of $V_{mem}$ with a positive exponent[77], $v_{ahp}$ represents a after-hypalarizing term that is increased with every output spike, and which has a negative feedback onto the membrane potential, typical of spike-frequency adaptation mechanisms[78]. Indeed, the term $\delta_{spk}(t)$ is 1 when the neuron spikes, and zero otherwise. The terms $\tau_{mem}$ and $\tau_{ahp}$ represent the time constants of the membrane potential and after-hypalarizing potential respectively (see Table 4 for the values used in the simulations of the neural architecture). The term $I_{syn}(t)$ represents the total weighted sum of the synaptic input, which in our network is composed of one excitatory and one inhibitory synaptic input that are subtracted from each other ($I_{syn}(t) = I_{exc}(t) - I_{inh}(t)$). The equations that govern the dynamics of the synaptic excitatory and inhibitory circuits are, to first order approximation:

$$\tau_{exc}\frac{d}{dt}I_{exc}(t) + I_{exc}(t) = w_{exc}\delta_{UP}(t)$$
$$\tau_{inh}\frac{d}{dt}I_{inh}(t) + I_{inh}(t) = w_{inh}\delta_{DN}(t)$$

(2)

where $\tau_{exc}$ and $\tau_{inh}$ represent the time constants of the synapses, $w_{exc}$ and $w_{inh}$ their weights (see also Table 4 for the values used in the simulations). The terms $\delta_{UP}(t)$ and $\delta_{DN}(t)$ are one during UP and DN input spikes respectively, and zero otherwise.

**SNN model signal processing features.** The software simulations made in the prototyping stage to model the hardware implementation of the Spiking Neural Network (SNN) used equations derived from the Differential Pair Integrator (DPI) synapse circuit analysis[13,59], by taking into account circuit constraints, such as 20% variability in all state variables due to device mismatch, or the fact that all variables

**Table 4 Synapse parameters of the SNN model depicted in (Fig. 1d).**

| Parameter | Value | Description |
|---|---|---|
| $\tau_{mem}$ | 15.2 ms | Time constant of the neuron's membrane potential. A value of 3.5 pA is set to the bias $I_{\tau}$ to achieve this time constant. |
| $\tau_{ahp}$ | 35.7 ms | Time constant of the neuron's after-hyperpolarizing potential. A value of 1 pA is set to the bias $I_{\tau_{ahp}}$ to achieve this time constant. |
| $\tau_{exc}$ | min: 3 ms max: 6 ms | Time constant of the excitatory synapses. This time constant determines the cutoff frequency of the DPIs receiving the UP spikes. To get this these values we set the bias $I_{\tau_{exc}}$ in the range 8.9-17.8 pA |
| $\tau_{inh}$ | min: 2 ms max: 5.7 ms | Time constant of the inhibitory synapses. This time constant determines the cutoff frequency of the DPI receiving the DN spikes. To get this these values we set the bias $I_{\tau_{inh}}$ in the range 9.4-26.8 pA |
| $w_{exc}$ | 1 or 2 nA | With these weight values and the parameters mentioned above, the silicon neurons need a |
| $w_{inh}$ | −1 or −2 nA | minimum of 14 input excitatory spikes at a rate of 3kHz to generate a single spike. |

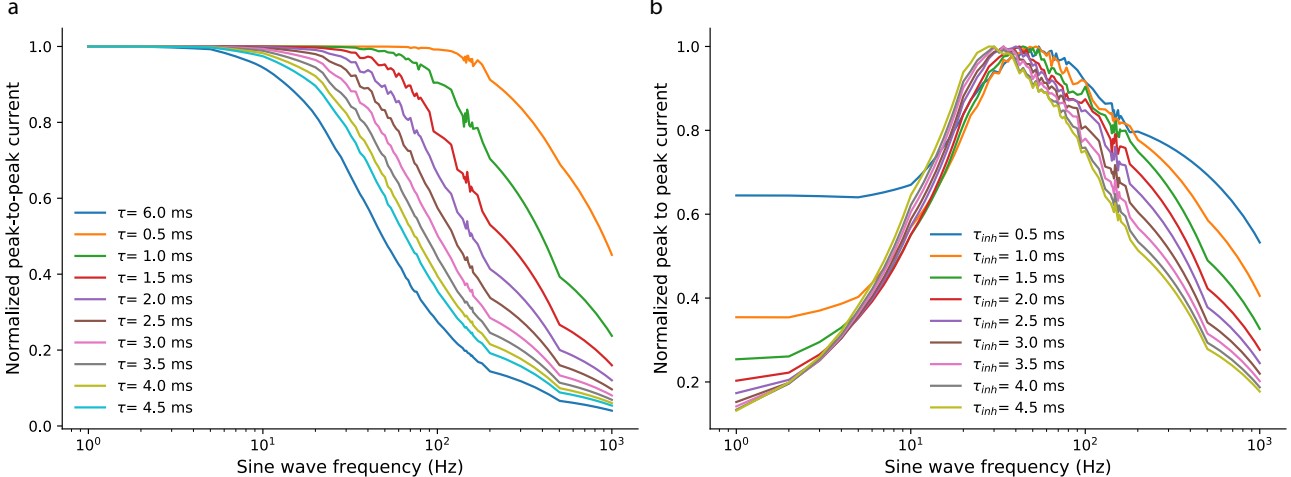

**Fig. 6 DPI low- and band-pass filter characteristics for spiking inputs. a** Behavioral simulation results for the normalized steady-state response of the Differential Pair Integrator (DPI) to spike trains encoding an input sine wave, as a function of sine wave frequency. The DPI is able to reproduce a standard low-pass filter behavior for spiking inputs. **b** Band-pass filters resulting from the combination of DPIs with different time constants. A first-order band-pass filter results from subtracting the time responses to sine waves of varying frequencies of a single excitatory DPI synapse with a given time constant with the time responses of an inhibitory DPI synapse with a different time constant. The band-pass filters depicted here were obtained by using an excitatory DPI with a time constant of 6 ms and subtracting the activity of inhibitory DPIs with time constants ranging from 0.5–4.5 ms.

encoded by currents were clipped at zero (currents in the neuron and synapse circuits can only flow in one direction). Figure. 6a shows the behavioral simulation results for the normalized steady-state response of the DPI to spike trains encoding an input sine wave, as a function of sine wave frequency. As expected, the DPI reproduces a standard low-pass filter behavior, also for spiking inputs. By combining the response of the excitatory DPI synapse with the one of the inhibitory synapse and appropriately choosing their time constants, we effectively designed a band-pass filter coarsely tuned to the spectral properties of High Frequency Oscillations (HFOs) (see Fig. 6b). Then, by exploiting the device mismatch effects in the synapse and neuron circuits (also simulated in software) and combining the output of multiple, slightly different neurons, we created an ensemble of "weak classifiers" that can, collectively, detect the occurrence of an HFOs in the data and to generalize to the slight variations present in the HFO signals.

**Software simulation and hardware validation of the neural architecture**. For the software simulation of the network we used the Spiking Neural Network simulator Brian2[79] and a custom made toolbox[58] that makes use of equations which describe the behavior of the neuromorphic circuits. To find the optimal parameters of the SNN, we were guided by the clinically relevant HFO marked by the Morphology Detector[31]: Around the HFO marked in the iEEG[31] we created snippets of data ± 25 ms. These snippets were used to select the parameters for the ADM and the SNN network (see Methods). The SNN architecture was validated using the previous generation of the neuromorphic processor DYNAP-SE[47], for which a working prototyping framework is available. The high-level software-hardware interface used to send signals to the SNN, configure its parameters, and measure its output was designed in collaboration with SynSense AG, Switzerland.

**Patient data**. We analyzed long-term iEEG recordings from the medial temporal lobe of 9 patients. Patients had drug-resistant focal epilepsy as detailed in Table 1. Presurgical diagnostic workup at Schweizerische Epilepsie-Klinik included

recording of iEEG from the medial temporal lobe. The independent ethics committee approved the use of the recorded data for research and patients signed informed consent. The surgical planning was independent of HFO. Patients underwent resective epilepsy surgery at UniversitätsSpital Zürich. After surgery, the patients were followed-up for >1 year. Postsurgical outcome was classified according to the International League Against Epilepsy (ILAE)[80].

The data set is publicly available[43] and is considered a standard dataset for HFO benchmarking[36] as it fulfills the following requirements[29].

- iEEG sampling rate of 2000 Hz
- low thermal noise level of the iEEG (<30 nV/$\sqrt{Hz}$)
- iEEG recorded during periods of slow-wave sleep, which promotes low muscle activity and high HFO rates
- several intervals from the same patient recorded during the same night and subsequent nights for test-retest analysis
- intervals are at least three hours apart from epileptic seizures to eliminate the influence of seizure activity
- after iEEG recording, patients underwent epilepsy surgery where a volume of the brain was resected
- documentation of the electrode contacts that were localized in the resected brain volume
- documentation of post-operative seizures, i.e. whether seizure freedom was achieved for >1 year

The amount of nights and intervals varied across patients with up to six intervals (≈5 min each) that were recorded in the same night (Table 1). We focused on the 3 most medial bipolar channels from recordings in the medial temporal lobe because HFO in these channels are known to have higher signal-to-noise ratio[31]. In total, we analyzed 18 hours of data recorded from 206 electrode contacts. In a previous publication with this data set, we had detected 34,479 HFO with the Morphology detector[31,37] and compared the HFO area to the resected brain volume to predict seizure outcome in order to validate the HFO detection[31].

**HFO detection**. HFO detection was performed independently for each channel in each 5-min interval of iEEG. The signal pre-processing steps consisted of bandpass filtering, baseline detection and transforming the continuous signal into spikes using the ADM block. The ADM principle of operation is as follows: whenever the amplitude variation of the input waveform exceeds an upper threshold $V_{tu}$ a positive spike on the UP channel is generated; if the change in the amplitude is lower than a threshold $V_{td}$, a negative spike in the DN channel is produced.

As the amplitude of the recordings changed dramatically with electrode, patient data and recording session, we introduced a baseline detection mechanisms that was used to adapt the values of the $V_{tu}$ and $V_{td}$ thresholds in order to produce the optimal number of spikes required for detecting HFO signals while suppressing the background noise and outliers in the recordings. This baseline was calculated for each iEEG channel in software: during the first second of recording, the maximum signal amplitude was computed over non-overlapping windows of 50 ms. These values were then sorted and the baseline value was set to the average of the lowest quartile. This procedure excluded outliers on one hand, and suppressed the noise floor on the other hand. This procedure was optimal for converting the recorded signals into spikes.

Spikes entered the SNN architecture as depicted in Fig. 5b, c. The SNN parameters used to maximize HFO detection were selected by analyzing the Inter-Spike-Intervals (ISIs) of the spike trains produced by the ADM and comparing their characteristics in response to inputs that included an HFO event versus inputs that had no HFO events. This analysis was then used to tune the time constants of the SNN output layer neurons and synapses. Specifically, the approach used was to rely on an ensemble of neurons in the output layer and to tune them with parameters sampled from a uniform distribution. The average time constant for the neurons was chosen to be 15 ms, with a coefficient of variation set by the analog circuit devise mismatch characteristics, to approximately 20%. Similarly, the excitatory synapse time constants were set in the range (3–6) ms and the inhibitory synapse time constants in the range (0.1–1) ms.

After sending the spikes produced by the ADMs to the SNN configured in this way, we evaluated snippets of 15 milliseconds output data produced by the SNN and signaled the detection of an HFO every time spikes were present in consecutive snippets of data. Outlier neurons in the hardware SNN that spiked continuously were considered uninformative and were switched off for the whole study. The activity of the rest of the neurons faithfully signaled the detection of HFO (see Table 2). For the HFO count, spikes with inter-spike-intervals <15 ms were aggregated to mark a single HFO.

**Post-surgical outcome prediction**. To retrospectively "predict" the postsurgical outcome of each patient in this data set, we first detected the HFO in each 5-min interval by measuring the activity of the silicon neurons in the hardware SNN. We calculated the rate of HFO per recording channel by dividing the number of HFO in the specific channel by the duration of the interval. The distribution of HFO rates over the list of channel defines the HFO vector. In this way we calculated an HFO vector for each interval in each night. We quantified the test-retest reliability of the distribution of HFO rates over intervals by computing the scalar product of all pairs of HFO vectors across intervals (Table 1). We then delineated the "HFO area" by comparing the average HFO rate over all recordings and choosing the area at the electrodes with HFO rates exceeding the 95 percentile of the rate distribution. Finally, to assess the accuracy of the patient outcome prediction, we compared the HFO area identified by our procedure with the area that was resected in surgery, and compared it with the postsurgical seizure outcome (Table 1).

**Ethics statement**. The study was approved by the institutional review board (Kantonale Ethikkommission Zurich PB-2016-02055). All patients signed written informed consent. The study was performed in accordance with all relevant ethical guidelines and regulations.

## Data availability

The iEEG data analyzed here are freely available at OpenNeuro in BIDS format under https://openneuro.org/datasets/ds003498.

## Code availability

The code used in this study is available at https://github.com/kburel/SNN_HFO_iEEG (https://zenodo.org/badge/latestdoi/359535894).

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

## Acknowledgements

This project has received funding from Swiss National Science Foundation (SNSF 320030_176222) and from the European Research Council (ERC) under the European Union's Horizon 2020 research and innovation program grant agreement No. 724295. We thank our colleagues Ole Richter and Ning Qiao for the design of the SNN cores on the chip presented, as well as the design of the spike encoders and asynchronous buffers in the analog headstage. We are also grateful to Dmitrii Zendirkov for support in measuring the variability of the synapse and neuron circuits, and Chenxi Wu, Carsten Nielsen and Adrian M. Whatley, for developing the software and firmware required to communicate with the hardware platform and for configuring the parameters on chip.

## Author contributions

J.S. and G.I. conceived the experiments, M.S. and K.B. conducted the experiments and analyzed the results. All authors wrote and reviewed the manuscript.

## Competing interests

The authors declare no competing interests.
