## [Peer Review File · Nature Communications]

Reviewers' Comments:

Reviewer #1:

Remarks to the Author:

This paper presents a high-frequency oscillation detection architecture as well as its neuromorphic circuit implementation. They first combined a neural recording headstage with a signal-to-spike conversion circuit and a SNN implementation on the same chip for iEEG recordings. Although it is with some interesting ideas, it cannot be applied to a wider field of the community. Neither this study has solved the substantial problems in the field of neuromorphic engineering, nor it has proposed novel methodology for artificial intelligent applications. The contributions of this study is limited and the ideas for this study are not significantly novel. However, the solutions and the implementation methods for iEEG are of considerable importance, and this study is a good work in the field of analog implementation of SNNs. The authors should further highlight its contribution and significance in the field of neuromorphic computing, even in the field of brain-inspired computing if possible. The novelty of this study should also be classified, because combine recording, processing with detecting functions in a single hardware cannot be regarded as a highly novel work, especially in the publication Nature Communications.

Reviewer #2:

Remarks to the Author:

The paper is original as it constitutes a pilot real time application of a SNN model and a neuromorphic chip for HFO detection related to precise location of epileptic events.

This is a true demonstration of the advantages of neuromorphic computation for real time medical applications.

The paper proposes a new method and offers a neuromorphic chip that are tested on the detection of HFO of 9 patients in Zurich hospital. The presented results are convincing and promising.

There are some points that would still need some clarification:

- How are the locations of the implants decided before data is collected and how important the exact location is on the results.
- Why the SNN model has 256 fixed hidden nodes, rather than an evolving layer as already suggested in some referenced publications?
- Would the fixed number of neurons be a limitation for a further use of the chip?
- How exactly the the UP-spike and the DOWN-spike inputs are integrated into the SNN and is it biologically plausible?
- Along with the many advantages the proposed system offers, there are some limitations that need to be discussed.

Reviewer #3:

Remarks to the Author:

The major claim of the paper is to propose and design a neuromorphic system that combines for the first time a neural recording headstage with a signal-to-spike conversion circuit and a multi-core spiking neural network (SNN) architecture on the same die for recording, processing, and detecting clinically relevant HFOs in iEEG from epilepsy patients. This is novel and interesting to researchers in the community. The conclusion is original and it will influence thinking in the field.

The following revisions are suggested to improve the paper:

- 1)The low sensitivity of 33% should be analyzed and discussed. Possible improvement methods can be given.
- 2)It is better to test the system on animals or human to verify

that the system can real-time detect HFO areas directly in the operation room.

3)The comparisons with other non-neuromorphic detections of HFO could be given with adv./disadv. This will give readers a more complete picture.

Response to Reviewers

An electronic neuromorphic system for real-time detection of High Frequency Oscillations (HFOs) in intracranial EEG

Dear Editor,

We thank the reviewers for taking the time to review the paper and their comments that have helped us to improve the manuscript.

Please find our point-by-point replies below.

On behalf of all authors,

Giacomo Indiveri

Reviewer 1

Comment 1.1: *This paper presents a high-frequency oscillation detection architecture as well as its neuromorphic circuit implementation. They first combined a neural recording headstage with a signal-to-spike conversion circuit and a SNN implementation on the same chip for iEEG recordings. Although it is with some interesting ideas, it cannot be applied to a wider field of the community. Neither this study has solved the substantial problems in the field of neuromorphic engineering, nor it has proposed novel methodology for artificial intelligent applications. The contributions of this study is limited and the ideas for this study are not significantly novel. However, the solutions and the implementation methods for iEEG are of considerable importance, and this study is a good work in the field of analog implementation of SNNs.*

Reply 1.1: We thank the Reviewer for these comments. We agree with the reviewer that the contribution of this study is on a very specific neuromorphic use case of HFO detection and it is not a general solution that can be applied to a wider field of neuromorphic computing. The main focus of the paper is in demonstrating the ability to carry out on-line HFO detection of iEEG signals with SNNs implemented using noisy mixed-signal circuits that are very low power. However, the approach can be extended to more general bio-signal processing problems similar to the ones cited in the paper (e.g., see references^{10,11} in the main manuscript). Besides these approaches that apply neuromorphic methods to biomedical signals, we compared our work also to other studies that apply processing methods ranging from conventional digital signal processing to high dimensional vector data processing methods⁵³⁻⁵⁶. However, unlike these studies, our work proposes the application of a neuromorphic processor that was designed from the onset with an additional analog headstage for processing bio-signals signals in real-time. This hybrid signal processing methodology is quite unique and combines on the same die analog pre-processing with neuromorphic post-processing: amplification, band-pass filtering, and signal-to-spike conversion is done in the analog pre-processing stage while spike-based pattern recognition is performed using the neural processing cores. As opposed to other state-of-the-art HFO detection methods (e.g., see²³), our system can perform in real-time. The signal-to-spike encoding method used, although not novel, exploits a class of data analysis referred to as “symbolization”, which describes the process of transforming experimental measurements into a series of discrete symbols. Symbolization is particularly interesting for iEEG analysis because it faithfully preserves dominant dynamical signal characteristics while significantly increasing the efficiency of detecting and quantifying information contained in real-world time series⁵³. Compared to synchronous sampling and symbolization used by other state-of-the-art approaches⁵³, the encoding method implemented in this paper is ideally suited for the neuromorphic processing core, in the sense that time represents itself and there is no periodic check for signal dynamics. Rather, the signal itself determines the dynamics of the encoding.

So, we concur with the reviewer that the paper does not propose a novel methodology for artificial intelligent applications. However, we believe that the on-line detection of HFOs using a compact embedded device has the potential of leading to the development of an embedded system that solves a clinically relevant problem: to support the detection of the epileptogenic zone in the operating room, during surgical operations.

Action: We now added a new a paragraph to specify how the contribution of the paper is limited to the specific HFO detection application and address these comment in the Discussion Section at lines 260–264.

Comment 1.2: *The authors should further highlight its contribution and significance in the field of neuromorphic computing, even in the field of brain-inspired computing if possible. The novelty of this study should also be classified, because combine*

recording, processing with detecting functions in a single hardware cannot be regarded as a highly novel work, especially in the publication Nature Communications.

Reply 1.2: We thank the Reviewer for this encouragement. Given that the HFO detection network uses the same circuits present in the DYNAP-SE architecture³⁷, we refer the reader to that publication³⁷ for the significance of this approach to the neuromorphic computing field in general. However to address this comment we added the following material to better compare the system proposed to other related state-of-the-art approaches:

Action 1: We added a paragraph to highlight the significance of the work proposed and further emphasized the novelty of the work with respect to the HFO detection (lines 304–323).

“When comparing our neuromorphic SNN with other HFO detectors in the literature²⁹, there are several differences and also commonalities. As a conceptual difference, the SNN closely follows computational principles of biological neurons and neuronal networks. This translates into very low power consumption and extended local processing. The SNN’s output is condensed to the binary signalling of either the presence or absence of an HFO. As an advantage, the low power consumption will benefit future applications of the SNN in implantable devices for long-term monitoring of the epilepsy severity. As a design choice, the raw signal is not stored and post-hoc examination of HFOs is not possible, e.g. for distinguishing HFOs from artifact interference. Therefore, our stand-alone HFO detector must provide reliable information to the surgeon with sufficient sensitivity and specificity. As a commonality, there are some non-neuromorphic HFO detectors^{18,23–26,33} that also validate their HFOs to “predict” seizure freedom after resective epilepsy surgery in individual patients (Table 1). In the “prediction” across the patient group (Table 2), the sensitivity stands out to be very low, i.e. even after the HFO area was resected, Patients 7 and 9 suffered from recurrent seizures (FN prediction). On one hand, a FN may be due to insufficient HFO detection. On the other hand, the spatial sampling of iEEG recordings may be insufficient for localizing the EZ, i.e. seizures may originate from areas where they remain undetected by the iEEG recordings. This spatial sampling restriction affects the two HFO detectors and current clinical practice all alike. Indeed, current clinical practice advised resection of brain tissue in Patients 7, 8, and 9, which nevertheless suffered from recurrent seizures postoperatively, i.e. the EZ was not removed in its entirety. Interestingly, the HFO area of the Hardware SNN might have included the correct EZ in Patient 8 (TP) while this was not the case for the Morphology detector (FN). Still, the NPV in Table 2 is the most relevant quantity for clinical purpose and it is sufficiently high for both the Morphology Detector and the Hardware SNN.”

Action 2: We created a new Table to highlight the achievements of this work compared to state-of-the-art systems. This Table lists the four most recent devices that aim for low-power bio-signal processing, and highlights how the approach we propose is the most efficient in terms of power consumption (see lines 271–284 and Table 3 in the manuscript).

“Other embedded systems and VLSI devices designed for the specific case of processing and/or classifying EEG signals have been proposed in recent years^{53–56}. Table 3 highlights the differences between these systems and the one presented in this work. Interestingly, only two of these other designs have opted for integrating analog acquisition headstages with the computing stages for standalone operation^{55,56}. Both of these designs have a comparable number of channels to our system; however they comprise conventional analog to digital converter designs (ADCs) that are not optimal for processing bio-signals⁵⁷. Indeed, analog to digital data conversion for bio-signal processing has been an active area of investigation in biomedical processing field, with increasing evidence in favor of delta encoding schemes (such as the one used in this work)^{58–60}. The only design listed in Table 3 that utilizes a symbolic encoding scheme and that has been applied to iEEG is the work of Burello et al.’19⁵³. However, that design is missing a co-integrated analog headstage and, by extension, an integrated local binary pattern encoder. Separating the signal encoding stage from the processing stage allows the implementation of sophisticated signal processing techniques and machine learning algorithms, as is evident from the works of Burello et al.’19⁵³ and Feng et al.’17⁵⁴. But using off-the shelf platforms for signal encoding, processing, or both, leads to much higher power consumption and bulky platforms that make the design of compact and portable embedded systems more challenging.”

Response to the Reviewers' comments

Feature/Design	This work	Burello'19 ⁵³	Feng'17 ⁵⁴	Van Helleputte'14 ⁵⁵	Yoo'12 ⁵⁶
Analog Headstage	8ch+32filters	No	No	3Ch ECG+ETI	8Ch+8filters
Encoding	Asynchronous delta-encoding	Synchronous local binary pattern	External ADC @173.61S/s/@256S/s	Sigma-delta ADC @500S/s	Multiplexed SAR ADC @32KS/s
Application	HFO detection	Seizure detection	Seizure detection	Personal health monitoring	Seizure classification
Power consumption (headstage+processor)	58.4uW+555.6uW	64mW*	45mW***	183uW+191uW**	66uW + 2.03uJ/classification
Platform	Custom analog + DYNAP	Nvidia Tegra X2	Altera Cyclone II FPGA	Custom analog+ ARM CortexM0	Custom
Learning/Processing	SNN, randomized dataset	Hyperdimensional vectors, specific dataset	SVM, both randomized and specific datasets	ICA, PCA, CWT and feature extraction	SVM, specific dataset

* Power for processing 24 channels. ** Power measured for R-peak detection with motion artefact reduction. *** Not multichannel.

Reviewer 2

Comment 2.1: *The paper is original as it constitutes a pilot real time application of a SNN model and a neuromorphic chip for HFO detection related to precise location of epileptic events. This is a true demonstration of the advantages of neuromorphic computation for real time medical applications. The paper proposes a new method and offers a neuromorphic chip that are tested on the detection of HFO of 9 patients in Zurich hospital. The presented results are convincing and promising.*

Reply 2.1: We thank the Reviewer for this positive judgment.

Comment 2.2: *There are some points that would still need some clarification: How are the locations of the implants decided before data is collected and how important the exact location is on the results.*

Reply 2.2: The locations of the depth electrodes are planned depending on the pre-surgical workup, which includes the analysis of seizure etiology, MR imaging, and scalp EEG. There is a trade-off between patient safety (i.e. less electrodes) and complete spatial sampling of the brain (i.e. more electrodes). Unfortunately, there are patients where the epileptogenic zone is not completely sampled, which may lead to suboptimal surgical planning. In our analysis, the incomplete sampling of the epileptogenic zone may result in "False Negative (FN)" where the HFO area was fully located inside the resection volume but who suffered from recurrent seizures. This incomplete sampling may have occurred in Patients 7 and 9 (Table 1). However, the limited spatial sampling is inherent to all invasive recordings and independent of the HFO detector used.

Action: We have now added a short explanation where we describe the FN patients (lines 251–253).

"The false prediction may stem either from HFOs being insufficiently detected or from the epileptogenic zone being insufficiently covered by iEEG electrode contacts."

In a wider context, we have entered the following phrases in discussion section (lines 304–323):

"In the "prediction" across the patient group (Table 2), the sensitivity stands out to be very low, i.e. even after the HFO area was resected, Patients 7 and 9 suffered from recurrent seizures (FN prediction). On one hand, a FN may be due to insufficient HFO detection. On the other hand, the spatial sampling of iEEG recordings may be insufficient for localizing the EZ, i.e. seizures may originate from areas where they remain undetected by the iEEG recordings. This spatial sampling restriction affects the two HFO detectors and current clinical practice all alike. Indeed, current clinical practice advised resection of brain tissue in Patients 7, 8, and 9, which nevertheless suffered from recurrent seizures postoperatively, i.e. the EZ was not removed in its entirety.

Interestingly, the HFO area of the Hardware SNN might have included the correct EZ in Patient 8 (TP) while this was not the case for the Morphology detector (FN). Still, the NPV in Table 2 is the most relevant quantity for clinical purpose and it is sufficiently high for both the Morphology Detector and the Hardware SNN.”

Comment 2.3: *Why the SNN model has 256 fixed hidden nodes, rather than an evolving layer as already suggested in some referenced publications?*

Reply 2.3: We used a full core of the neuromorphic processor to detect the HFO signals with all the 256 neurons in the core (the processor comprises 4 cores of 256 neurons each that share the same set of biases³⁷). However, thanks to this comment, we made a more thorough analysis of the results and found that an average number of 64 neurons are sufficient for detecting an HFO from a single channel input. In this study, we focus on recordings from a single channel, however, according to this analysis the method proposed should allow 4 channels per core to be processed in parallel, leading to 16 channels/chip. We are grateful to the reviewer for having raised this point that prompted us to push the methodology to the limit.

Action: We added a sentence in the results to comment on the average number of neurons required by the SNN proposed to detect HFOs (lines 210–211).

“Analysis of the data presented in Fig. 5 shows that an average number of 64 neurons was sufficient for detecting an HFO from a single channel input.”

Comment 2.4: *Would the fixed number of neurons be a limitation for a further use of the chip?*

Reply 2.4: The device presented is represents an academic feasibility study that uses the minimum amount of resource necessary to explore the processor’s computational abilities (i.e., 4 cores of 256 neurons each). A future product based on this study could easily extend the number of cores and increase the number of available neurons (for example the IBM TrueNorth chip has 4096 cores of 256 neurons, with $\approx 10^6$ neurons/chip.) We envisage that a future system of this type should be able to process 32 iEEG channels in parallel, which, according to the analysis above, would require 8 cores in the best case scenario and 32 cores in the worst case scenario.

Comment 2.5: *How exactly the the UP-spike and the DOWN-spike inputs are integrated into the SNN and is it biologically plausible?*

Reply 2.5: The UP-DOWN spike inputs are derived from a Delta-encoding scheme of the input signal. They are integrated into the SNN by sending them to excitatory and inhibitory synapses at each neuron. This is a heuristic transformation of the input time series that is inspired from sensory neurons that do spike rate coding. Our analysis that led to specifying the SNN network architecture showed that it works well in the neuromorphic system, too.

More specifically, each input spike to a synapse produces an exponentially decaying Excitatory or Inhibitory Post-Synaptic Current that is then integrated on the neuron’s membrane capacitor. Once (or if) the integrated signal reaches the neuron’s firing threshold, an output spike results⁷.

Comment 2.6: *Along with the many advantages the proposed system offers, there are some limitations that need to be discussed.*

Reply 2.6: We thank the Reviewer for encouraging us to extend our Discussion section.

Action: We now specify how the approach proposed is limited to a specific application domain, rather than being a general purpose neuromorphic processor solution (see lines 260–264) and highlight the advantages and disadvantages of the SNN compared to other non-neuromorphic HFO detectors in wider extent (lines 304–323):

“When comparing our neuromorphic SNN with other HFO detectors in the literature²⁹, there are several differences and also commonalities. As a conceptual difference, the SNN closely follows computational principles of biological neurons and neuronal networks. This translates into very low power consumption and extended local processing. The SNN’s output is condensed to the binary signalling of either the presence or absence of an HFO. As an advantage, the low power consumption will benefit future applications of the SNN in

implantable devices for long-term monitoring of the epilepsy severity. As a design choice, the raw signal is not stored and post-hoc examination of HFOs is not possible, e.g. for distinguishing HFOs from artifact interference. Therefore, our stand-alone HFO detector must provide reliable information to the surgeon with sufficient sensitivity and specificity. As a commonality, there are some non-neuromorphic HFO detectors^{18,23–26,33} that also validate their HFOs to “predict” seizure freedom after resective epilepsy surgery in individual patients (Table 1). In the “prediction” across the patient group (Table 2), the sensitivity stands out to be very low, i.e. even after the HFO area was resected, Patients 7 and 9 suffered from recurrent seizures (FN prediction). On one hand, a FN may be due to insufficient HFO detection. On the other hand, the spatial sampling of iEEG recordings may be insufficient for localizing the EZ, i.e. seizures may originate from areas where they remain undetected by the iEEG recordings. This spatial sampling restriction affects the two HFO detectors and current clinical practice all alike. Indeed, current clinical practice advised resection of brain tissue in Patients 7, 8, and 9, which nevertheless suffered from recurrent seizures postoperatively, i.e. the EZ was not removed in its entirety. Interestingly, the HFO area of the Hardware SNN might have included the correct EZ in Patient 8 (TP) while this was not the case for the Morphology detector (FN). Still, the NPV in Table 2 is the most relevant quantity for clinical purpose and it is sufficiently high for both the Morphology Detector and the Hardware SNN.”

Reviewer 3

Comment 3.1: *The major claim of the paper is to propose and design a neuromorphic system that combines for the first time a neural recording headstage with a signal-to-spike conversion circuit and a multi-core spiking neural network (SNN) architecture on the same die for recording, processing, and detecting clinically relevant HFOs in iEEG from epilepsy patients. This is novel and interesting to researchers in the community. The conclusion is original and it will influence thinking in the field.*

Reply 3.1: We thank the Reviewer for this positive judgment.

Comment 3.2: *The low sensitivity of 33% should be analyzed and discussed. Possible improvement methods can be given.*

Reply 3.2: We agree with the Reviewer that the low sensitivity needs some more explanation. The sensitivity 33% results from 2/3 poor-outcome patients being classified as FN (Table 1). The false prediction may stem 1) from HFOs being insufficiently detected or 2) from the epileptogenic zone being insufficiently covered by iEEG electrode contacts. The locations of the iEEG electrodes are planned depending on the pre-surgical workup, which includes the analysis of seizure etiology, MR imaging, and scalp EEG. There is a trade-off between patient safety (i.e. less electrodes) and complete spatial sampling of the brain (i.e. more electrodes). Unfortunately, there are patients where the epileptogenic zone is not completely sampled, which may lead to suboptimal surgical planning. In our analysis, the incomplete sampling of the epileptogenic zone may result in “False Negative (FN)” where the HFO area fully located inside the resection volume but who suffered from recurrent seizures. This incomplete sampling may have occurred in Patients 7 and 9 of Table 1. However, the limited spatial sampling is inherent to all invasive recordings and independent of the HFO detector used.

Action 1: We have now added a short explanation where we describe the FN patients (lines 251–253).

“The false prediction may stem either from HFOs being insufficiently detected or from the epileptogenic zone being insufficiently covered by iEEG electrode contacts.”

Action 2: In a wider context, we have entered the following phrases in the Discussion section (lines 304–323):

“In the “prediction” across the patient group (Table 2), the sensitivity stands out to be very low, i.e. even after the HFO area was resected, Patients 7 and 9 suffered from recurrent seizures (FN prediction). On one hand, a FN may be due to insufficient HFO detection. On the other hand, the spatial sampling of iEEG recordings may be insufficient for localizing the EZ, i.e. seizures may originate from areas where they remain undetected by the iEEG recordings. This spatial sampling restriction affects the two HFO detectors and current clinical practice all alike. Indeed, current clinical practice advised resection of brain tissue in Patients 7, 8, and 9, which nevertheless suffered from recurrent seizures postoperatively, i.e. the EZ was not removed in its entirety. Interestingly, the HFO area of the Hardware SNN might have included the correct EZ in Patient 8 (TP) while this was not the case for the Morphology detector (FN). Still, the NPV in Table 2 is the most relevant quantity for clinical purpose and it is sufficiently high for both the Morphology Detector and the Hardware SNN.”

Comment 3.3: *It is better to test the system on animals or human to verify that the system can real-time detect HFO areas directly in the operation room.*

Reply 3.3: We certainly agree with the Reviewer that testing the system on electrophysiological data is mandatory. Testing in an animal model of epilepsy has the disadvantage that electrophysiological findings in animals do not easily translate to application in patients.

Therefore, our manuscript validates our HFO detector on patients' intracranial EEG that was recorded during the course of the standard clinical practice before a surgery is proposed to the patient.

As a further step towards HFO detection directly in the operating room, we have just recently simulated HFO detection in data recorded during surgery. This new study analyses a separate dataset and applies a slightly modified HFO detector so that we have described the results in a separate manuscript, recently submitted to Scientific Reports. As this manuscript is still in the review phase, we cannot cite it yet.

Action: This topic is mentioned as an outlook in the last phrase of the Conclusion section (lines 332–335):

"The general approach of building on the same chip sensors that can convert their outputs to spikes, and of interfacing spiking neural network circuits and systems on the same chip can lead to the development of a new type of "neuromorphic intelligence" sensory-processing devices for tasks that require closed-loop interaction with the environment in real-time, with low latency and low power budget."

Comment 3.4: *The comparisons with other non-neuromorphic detections of HFO could be given with advantages/disadvantages. This will give readers a more complete picture.*

Reply 3.4: We thank the Reviewer for encouraging us to extend our discussion of the literature. The literature on non-neuromorphic HFO detectors is vast²⁹. Since the aim of epilepsy surgery is seizure freedom²², also HFO detectors must be validated against seizure freedom. Further, only if a detector applies a prospective definition of a clinically relevant HFO can the results be generalized to an independent dataset. To our knowledge, these criteria are only fulfilled by a small number of studies^{18,23–26}. Of particular interest is the Morphology detector²³, which we have used for training our SNN detector and which we benchmark against on the on the same dataset³³ in Table 2.

Action: We now highlight the advantages and disadvantages of the SNN compared to other non-neuromorphic HFO detectors in wider extent in the discussion section (lines 304–323):

"When comparing our neuromorphic SNN with other HFO detectors in the literature²⁹, there are several differences and also commonalities. As a conceptual difference, the SNN closely follows computational principles of biological neurons and neuronal networks. This translates into very low power consumption and extended local processing. The SNN's output is condensed to the binary signalling of either the presence or absence of an HFO. As an advantage, the low power consumption will benefit future applications of the SNN in implantable devices for long-term monitoring of the epilepsy severity. As a design choice, the raw signal is not stored and post-hoc examination of HFOs is not possible, e.g. for distinguishing HFOs from artifact interference. Therefore, our stand-alone HFO detector must provide reliable information to the surgeon with sufficient sensitivity and specificity. As a commonality, there are some non-neuromorphic HFO detectors^{18,23–26,33} that also validate their HFOs to "predict" seizure freedom after resective epilepsy surgery in individual patients (Table 1). In the "prediction" across the patient group (Table 2), the sensitivity stands out to be very low, i.e. even after the HFO area was resected, Patients 7 and 9 suffered from recurrent seizures (FN prediction). On one hand, a FN may be due to insufficient HFO detection. On the other hand, the spatial sampling of iEEG recordings may be insufficient for localizing the EZ, i.e. seizures may originate from areas where they remain undetected by the iEEG recordings. This spatial sampling restriction affects the two HFO detectors and current clinical practice all alike. Indeed, current clinical practice advised resection of brain tissue in Patients 7, 8, and 9, which nevertheless suffered from recurrent seizures postoperatively, i.e. the EZ was not removed in its entirety. Interestingly, the HFO area of the Hardware SNN might have included the correct EZ in Patient 8 (TP) while this was not the case for the Morphology detector (FN). Still, the NPV in Table 2 is the most relevant quantity for clinical purpose and it is sufficiently high for both the Morphology Detector and the Hardware SNN."

Reviewers' Comments:

Reviewer #1:

Remarks to the Author:

Comments to the Author

The authors of the paper have proposed a neuromorphic system for biomedical applications. They have analyzed its performance in detail. Comprehensive research has been done and described. In the updated version, they have tried to revise the manuscript according to my comments. However, there are several remained comments on this work, which needs to be further addressed.

- The brief description is presented in the main text. However, please add the detailed mathematical description of the model used on the neuromorphic hardware.
 - It is obviously important to select a good and proper model for neuromorphic computing, so the next question is why this model in this paper is selected, not others?
 - Based on the first comment, is there any SNN model that has better performance than the selected model in this study to deal with online data analyzing? It is very important to clarify the novelty and contribution of this work.
 - Is there any other standard data set for EEG signals? A fair comparison can be obtained if using a standard data set for testing the performance. Although the clinical data of patients is more meaningful for scientific problem, a standard data set is considered to be better for the performance test of hardware.
 - How to determine the parameter values used in the SNN model that is implemented on the presented hardware?
 - How to deal with the difference of the computation results between the pre-simulation on software and the final implementation on hardware?
 - Since analog neuromorphic hardware has more noise than digital circuits, how to compute more precisely based on the selected SNN model?
 - Digital neuromorphic system is an important candidate in the field of neuromorphic intelligence. In comparison with analog or mixed neuromorphic hardware, digital hardware has the advantages of high precision, low design complexity and reconfigurability. There are some representative works on digital neuromorphic computing in recent three years, such as [1]-[4]. Study #1 presented a multigranular digital neuromorphic architecture for building large-scale brain, which is the first and the most representative digital neuromorphic computing work towards brain-like cognition with multiple brain areas. It is also potentially applied in biomedical fields, including EEG signal analyzing. Study #2 and Study #3 are aiming at artificial general intelligence, and are also powerful to be applied in real-time detection of biomedical signals. Study #4 presents a vital approach to build a biologically plausible and meaningful neuromorphic system. It can also be applied in online data analyzing with proper SNN models. Although analog/mixed neuromorphic hardware has advantages, please also comment the advantages of these digital neuromorphic studies and discusses the comparison between the propose work and these works.
[1]BiCoSS: Toward Large-Scale Cognition Brain With Multigranular Neuromorphic Architecture. IEEE transactions on neural networks and learning systems.
[2] Loihi: A neuromorphic manycore processor with on-chip learning. IEEE Micro.
[3] Towards artificial general intelligence with hybrid Tianjic chip architecture. Nature.
[4] Scalable digital neuromorphic architecture for large-scale biophysically meaningful neural network with multi-compartment neurons. IEEE transactions on neural networks and learning systems (Early Access).
-
- Since analog neuromorphic hardware has more noise than digital circuits, how to compute more precisely based on the selected SNN model?
 - How to guarantee the reproductability of this study? If the neuromorphic hardware is used in other conditions to analyze EEG signals, how to guarantee its effectiveness?
 - What is the substantial difficulty for a neuromorphic hardware to deal with the high frequency oscillations with 80-500 Hz for EEG data?

- What is the major scientific problem(s) this study has solved and focused on? How does this study solve these problems? It should be further clarified.

Reviewer #2:

Remarks to the Author:

The authors have replied to all my questions and the issues raised in my review have been addressed in the revised version.

I am happy with the revised version and the response.

Reviewer #3:

Remarks to the Author:

The authors give some explanations on the reviewer's questions and comments. Some revisions have also been made in the revised manuscript.

Response to Reviewers (second revision)

An electronic neuromorphic system for real-time detection of High Frequency Oscillations (HFOs) in intracranial EEG

Dear Editor,

We thank the reviewers for this second round of revisions that highlighted some important points which we believe have been properly addressed in this round. We are particularly grateful to reviewer 1, who made us realize that there were some important aspects missing from the paper, such as the detailed description of the model and a more in depth discussion of the difference of approaches between this work and other neuromorphic devices. Please find our point-by-point replies below.

On behalf of all authors,

Giacomo Indiveri

Reviewer 1

Comment 1.1: *The authors of the paper have proposed a neuromorphic system for biomedical applications. They have analyzed its performance in detail. Comprehensive research has been done and described. In the updated version, they have tried to revise the manuscript according to my comments. However, there are several remained comments on this work, which needs to be further addressed.*

The brief description is presented in the main text. However, please add the detailed mathematical description of the model used on the neuromorphic hardware.

Reply 1.1: We agree that the model was not fully described. Thanks for pointing this out.

Action: We added a more thorough description of the model including the mathematical description of the neuron and synapse behavior in Section 5.2 at lines 372–393.

Comment 1.2: *It is obviously important to select a good and proper model for neuromorphic computing, so the next question is why this model in this paper is selected, not others?*

Reply 1.2: We approached the problem of HFO detection more from a signal-processing point of view, rather than following a classical ANN approach. As a consequence we built the model from the bottom up by engineering its parameters (e.g. weights and time constants) to match the signals we wanted to detect, from first principles.

Action: We added a paragraph to explain why the model has the structure proposed, and describing the approach followed to choose such a model in Section 5.3 (see lines 395–407).

Comment 1.3: *Based on the first comment, is there any SNN model that has better performance than the selected model in this study to deal with online data analyzing? It is very important to clarify the novelty and contribution of this work.*

Reply 1.3: We would like to point out that the focus of the paper is on presenting a compact, low-power device that can be used “at the edge” in real-time without requiring bulky or power-hungry computing systems. It is indeed plausible that with unlimited resources (memory, power, time, and data-sets), a better model could be developed in principle. On the other hand, we show that our (resource-limited) on-line system is actually on par with the state-of-the-art solutions that have been proposed in the literature, that only work off-line and that did not attempt to minimize power, size, or memory resources. To our knowledge there are no other SNN models presented in the literature that can outperform the one we proposed in this work, for HFO detection, when these resource-limitations are taken into account.

Action: We added a paragraph to clarify the novelty of this approach in the Introduction (see lines 15–21) and emphasized the novelty also in the Discussion at lines 299–313.

Comment 1.4: *Is there any other standard data set for EEG signals? A fair comparison can be obtained if using a standard*

data set for testing the performance. Although the clinical data of patients is more meaningful for scientific problem, a standard data set is considered to be better for the performance test of hardware.

Reply 1.4: While there are multiple datasets on more general EEG signals, our proposed system has a very focused application: the detection of clinically relevant High Frequency Oscillations (HFOs) in the *intracranial* EEG (iEEG). HFOs have appreciable energy at least up to 500 Hz. Consequently, for a dataset to be acceptable for benchmarking, it must have a sampling rate ≥ 2000 Hz. It is a principal advantage of our proposed system that it can easily, by design, deal with signals in that frequency range. A most recent review on HFOs³⁵ lists the data sets that are currently available for download. The dataset that we use here⁴¹ is among these datasets in this review and is considered a standard dataset for HFO benchmarking³⁵.

Action: We now describe in more detail the criteria that have to be fulfilled in order that a dataset can be used for benchmarking (see lines 423–439):

“However, for detecting HFOs and validating the results against the prediction of post-operative seizure freedom, i.e. whether the detected HFOs are indeed clinically relevant, the following requirements must be fulfilled:

- iEEG sampling rate of 2000 Hz
- low thermal noise level of the iEEG ($< 30 \text{ nV}/\sqrt{\text{Hz}}$)
- iEEG recorded during periods of slow-wave sleep, which promotes low muscle activity and high HFO rates
- several intervals from the same patient recorded during the same night and subsequent nights for test-retest analysis
- intervals are at least three hours apart from epileptic seizures to eliminate the influence of seizure activity
- after iEEG recording, patients underwent epilepsy surgery where a volume of the brain was resected
- documentation of the electrode contacts that were localized in the resected brain volume
- documentation of post-operative seizures, i.e. whether seizure freedom was achieved for > 1 year

The dataset used in this work is the only of the available data sets³⁵ that fulfills all of the above criteria.”

Comment 1.5: *How to determine the parameter values used in the SNN model that is implemented on the presented hardware?*

Reply 1.5: The choice of the hyper-parameters of the network was done via behavioral software simulations in the setup prototyping stage. As this network was designed from first principles (e.g., by analyzing the spectral components of the signals and choosing the frequency bands of interest), we used the predictions made from theory to pick the time constants of the synapses (related to the cut-off frequencies of their equivalent filtering properties), and picked a distribution of such parameters in software that was consistent with the one predicted for the hardware, from the device mismatch estimates we measured from the chip (see Fig. 1).

Action: The new sections added to address the previous issues raised (Sections 5.2 and 5.3) also explain which parameters were tuned and the strategy used for hyper-parameter tuning. We further explain in the Discussion (lines 299–313) how the procedure used to find the model hyper-parameters differs from the ones used in deep-network and machine learning approaches.

Comment 1.6: *How to deal with the difference of the computation results between the pre-simulation on software and the final implementation on hardware?*

Reply 1.6: The software simulation takes into account the device mismatch effects by picking the synapse parameters randomly, from a distribution that has the same characteristics of the circuits simulated. After mapping the parameters to the hardware we could verify (with great satisfaction) that the hardware results were in very good accordance with the software simulation results.

The robustness of the software simulation has now been confirmed in an independent dataset of intraoperative ECoG recordings, where the exact same simulated SNN presented here highly successfully detected HFOs - without fine tuning of parameters at all (see reference³⁷ in the manuscript and it's pre-print at arXiv:2011.08783).

Action: This procedure is now clarified in the Discussion lines 324–326 and in Section 5.3.

Figure 1. Normalized distribution of excitatory synaptic time constants measured from 1000 different DPI circuits.

Comment 1.7: *Since analog neuromorphic hardware has more noise than digital circuits, how to compute more precisely based on the selected SNN model?*

Reply 1.7: As explained in the new sections added (Sections 5.2 and 5.3), the approach followed is that of combining multiple weak-classifiers following an “ensemble” classification approach. Theories of ensemble techniques (random forests, bagging, boosting, etc.) have been proven to be very powerful computationally. The device mismatch and noise present in the mixed-signal circuits adds indeed noise, and makes individual classifiers (a single-neuron Perceptron in this case) very “weak”. However, we can rely on the fact that we integrated on the chip 1024 of these weak classifiers, and the measurements made are consistent with these ensemble theories that show how classification error decreases exponentially with the number of classifiers in the ensemble. Indeed, for the purpose of this work we estimated that even only 64 neurons/weak-classifiers would be sufficient to achieve the performance figures reported.

Comment 1.8: *Digital neuromorphic system is an important candidate in the field of neuromorphic intelligence. In comparison with analog or mixed neuromorphic hardware, digital hardware has the advantages of high precision, low design complexity and reconfigurability. There are some representative works on digital neuromorphic computing in recent three years, such as⁵⁻⁸. Study number 1 presented a multigranular digital neuromorphic architecture for building large-scale brain, which is the first and the most representative digital neuromorphic computing work towards brain-like cognition with multiple brain areas. It is also potentially applied in biomedical fields, including EEG signal analyzing. Study number 2 and Study number 3 are aiming at artificial general intelligence, and are also powerful to be applied in real-time detection of biomedical signals. Study number 4 presents a vital approach to build a biologically plausible and meaningful neuromorphic system. It can also be applied in online data analyzing with proper SNN models. Although analog/mixed neuromorphic hardware has advantages, please also comment the advantages of these digital neuromorphic studies and discusses the comparison between the propose work and these works.*

[1]BiCoSS: Toward Large-Scale Cognition Brain With Multigranular Neuromorphic Architecture. *IEEE transactions on neural networks and learning systems*.

[2] Loihi: A neuromorphic manycore processor with on-chip learning. *IEEE Micro*.

[3] Towards artificial general intelligence with hybrid Tianjic chip architecture. *Nature*.

[4] Scalable digital neuromorphic architecture for large-scale biophysically meaningful neural network with multi-compartment neurons. *IEEE transactions on neural networks and learning systems*

Reply 1.8: We fully agree with the reviewer. We pointed this out in the introduction (lines 15–21) and explained in the Discussion (lines 299–313) how the approaches based on the large-scale digital neuromorphic systems mention differ from the approach proposed based on analog circuits. The approach presented in this work is not aimed at competing or outperforming the digital ones. Rather it can be used to complement them or be used as a processing stage to provide pre-processed inputs to them.

Action: We added citations to the papers mentioned at lines 15–21 and explained how our approach differs from the ones cited in the Discussion (lines 299–313).

Comment 1.9: *How to guarantee the reproductability of this study? If the neuromorphic hardware is used in other conditions to analyze EEG signals, how to guarantee its effectiveness?*

Reply 1.9: The software code used to simulate the network proposed is available upon request, and the dataset used is publicly available on-line. All circuit details of the analog head-stage front-end have been listed in the manuscript, as all details of the synapse, neuron, and parameters used for the SNN processing stage. The details of the SNN circuits on the other hand are publicly available in the manuscripts cited (e.g., see^{13,45}). Colleagues and peers can also get access to our current hardware upon request. So this very specific study should be reproducible by any interested party.

Reproducing our results in other datasets (and live measurements from patients) is indeed the major long-term goal of this study. Already in our manuscript, using pre-recorded iEEG, we found that HFO rates are reproducible over several nights (Fig. 1i) in several patients (Table 1). We are currently in the process of encapsulating the setup used in the prototyping stages in a compact and electrically isolated package, so that we can use it live in a the operating room. But this goes well beyond the scope of the study presented here.

Furthermore, we applied the same SNN model with the exact same network parameters on a dataset of intraoperative ECoG recordings. The study was accepted for publication just now³⁷. This demonstrates the robustness of our approach and that its success is reproducible in independent data.

Comment 1.10: *What is the substantial difficulty for a neuromorphic hardware to deal with the high frequency oscillations with 80-500 Hz for EEG data?*

Reply 1.10: With digital neuromorphic architectures, the main challenge lies in the interfacing and encoding of the analog wave forms with a spike-based representation that is compatible with the hardware, and the speed of the simulation, in order to be able to receive streaming data on-line continuously. With our analog circuits approach, "time represents itself", so the circuits have computing times that naturally synchronized with the real-time input signals. The main challenge with these analog circuits lies in finding the right set of time-constants in the synapse and neuron circuits to properly cover the frequency spectrum of the input data, as explained in Sections 5.2, 5.2, and 5.4. In the our device, synaptic time constants can range from few μ s to hundreds of ms, so they can indeed detect signatures in high parts of the frequency spectrum, even up to 500 Hz.

Comment 1.11: *What is the major scientific problem(s) this study has solved and focused on? How does this study solve these problems? It should be further clarified.*

Reply 1.11: The major problem addressed in this manuscript is that of detection of HFOs online, as the iEEG signals are recorded directly from the brain. The scientific and technological challenge solved is that of doing this detection in a robust and reliable way, while using circuits and devices that minimize to very small numbers (e.g. sub mW) power consumption, and (as a consequence) network size and spiking activity (i.e., address event traffic and bandwidth requirements). In solving this problem, this work is demonstrating two main novel facts:

1. simple feed-forward single-layer spiking neural networks can be tuned to detect HFOs effectively, by exploiting the dynamics of their synapses and the properties of ensemble techniques that rely on variability in the neuron and synapse properties
2. mixed-signal low-power neuromorphic hardware, comprising only a handful (e.g., 256) of neurons, can be used to solve (very specific) clinically relevant problems reliably and robustly.

Even though this specific study does not address general purpose neuromorphic computing classes of problems, and does not propose to extend the state-of-the-art in machine learning or deep-learning methods, the very same questions raised in this review process highlight how these two facts are surprising and unexpected. By definition, computers can be programmed to simulate any algorithm, so it is possible in principle to develop a CPU/GPU based system (or even digital SNN neuromorphic architecture) to reproduce the results presented, and perhaps improve the HFO detection rates by some small percentage. But the main focus of this paper was to present a demonstration of how much can be done, with so little, in terms of power, memory, area, resolution, precision, and number of neurons, in a real-time online setting .

Action: We added paragraphs in the Introduction (lines 15–21), in the Discussion (lines 299–313) and in the new Sections (5.2 and 5.3) to address this comment.

Reviewer 2

Comment 2.1: *The authors have replied to all my questions and the issues raised in my review have been addressed in the revised version. I am happy with the revised version and the response.*

Reply 2.1: We are grateful to the reviewer for the time used in the previous round to review the manuscript and are happy we could address all the issues raised.

Reviewer 3

Comment 3.1: *The authors give some explanations on the reviewer's questions and comments. Some revisions have also been made in the revised manuscript.*

Reply 3.1: We hope the previous round of revisions and this one now fully addresses all of the issues raised, and are sincerely grateful for the comments made that we believe improved the quality of the manuscript significantly.

Reviewers' Comments:

Reviewer #1:

Remarks to the Author:

I appreciate the revision of the authors. I am happy with the revised version and the response.